

# Reanalysis of a ten year record (2004-2013) of seasonal mass balances at Langenferner / Vedretta Lunga, Ortler-Alps, Italy

Stephan Peter Galos[1], Christoph Klug[2], Fabien Maussion[1], Federico Covi[1], Lindsey Nicholson[1], Lorenzo Rieg[2], Wolfgang Gurgiser[1], Thomas Mölg[3], and Georg Kaser[1]

[1]Institute of Atmospheric and Cryospheric Sciences, University of Innsbruck, Austria
[2]Institute of Geography, University of Innsbruck, Austria
[3]Climate System Research Group, Institute of Geography, Friedrich-Alexander-University Erlangen-Nürnberg (FAU), Germany

*Correspondence to:* Stephan P. Galos (stephan.galos@uibk.ac.at)

**Abstract.** Records of glacier mass balance represent important data in climate science and their uncertainties affect calculations of sea level rise and other societally relevant environmental projections. In order to reduce and quantify uncertainties in mass balance series obtained by direct glaciological measurements, we present a detailed reanalysis work-flow which was applied to the ten year record (2004 to 2013) of seasonal mass balance of Langenferner, a small glacier in the European Eastern Alps. The approach involves a methodological homogenization of available point values and the creation of pseudo-observations of point mass balance for years and locations without measurements by the application of a process-based model constrained by snow line observations. We examine the uncertainties related to the extrapolation of point data using a variety of methods, and consequently present a more rigorous uncertainty assessment than is usually reported in the literature.

Results reveal that the reanalyzed balance record considerably differs from the original one mainly for the first half of the observation period. For annual balances these misfits reach the order of >300 $\mathrm{kg\,m^{-2}}$ and could primarily be attributed to a lack of measurements in the upper glacier part and to the use of outdated glacier outlines. For winter balances respective differences are smaller (up to 233 $\mathrm{kg\,m^{-2}}$) and they originate primarily from methodological inhomogeneities in the original series. Remaining random uncertainties in the reanalized series are mainly determined by the extrapolation of point data to the glacier scale and are in the order of $\pm 80$ $\mathrm{kg\,m^{-2}}$ for annual and $\pm 52$ $\mathrm{kg\,m^{-2}}$ for winter balances with values for single years / seasons reaching $\pm 136$ $\mathrm{kg\,m^{-2}}$. A comparison of the glaciological results to those obtained by the geodetic method for the period 2005 to 2013 based on airborne laser scanning data, reveals that no significant bias of the reanalyzed record is detectable.

## 1 Introduction

Long term records of glacier mass balance are of particular interest to the scientific community as they reflect the most direct link between observed glacier changes and the underlying atmospheric drivers (e.g., Hoinkes et al., 1967; Dyurgerov and Meier, 2000; Kaser et al., 2006; Mölg et al., 2009a). During the past decades, considerable effort has been made to establish programs which provide mass balance information of individual glaciers from all over the world (e.g., Zemp et al., 2009; WGMS, 2015).





These records are undoubtedly valuable, and, among others, form the basis for the assessment of glacier contribution to current sea level rise (Church et al., 2013). However, their usefulness is bounded by inhomogeneities and unquantified uncertainties in the limited number of available records (e.g., Cogley, 2009; Gardner et al., 2013; Zemp et al., 2009, 2015).

During recent years a number of studies have addressed the topic of inhomogeneous, biased and erroneous glacier mass balance records (e.g., Rolstad et al., 2009; Koblet et al., 2010; Zemp et al., 2010; Andreassen et al., 2012). While the number of measurement points needed to derive glacier-wide mass balance was discussed by Fountain and Vecchia (1999) and Pelto (2000), Jansson (1999) investigated the uncertainties related to direct measurement techniques and the appropriate number of measurement points at Storglaciären. Several studies have attempted to quantify how different methods of extrapolating point measurements to the glacier scale affect the resultant glacier mass balance (e.g., Funk et al., 1997; Hock and Jensen, 1999; Sold et al., 2016), while others focused on statistical approaches to evaluate annual or seasonal glacier mass balance and their associated confidence level (e.g., Lliboutry, 1974; Thibert and Vincent, 2009; Eckert et al., 2011). Zemp et al. (2013) provided a general concept for reanalyzing glacier mass balance series in the context of comparisons between directly measured and geodetically derived mass balance, which is commonly undertaken to cross-check the in-situ glaciological data (e.g., Funk et al., 1997; Østrem and Haakensen, 1999; Cox and March, 2004; Thibert et al., 2008; Cogley, 2009).

In addition, the use of subsidiary tools, such as distributed surface mass balance models, extensive accumulation measurements and auxiliary imagery have become more commonly used components of mass balance (re-)analyses. Mass balance models have been used to extrapolate point measurements to the glacier scale (e.g., Huss et al., 2009, 2013; Sold et al., 2016), to homogenize annual or seasonal mass balance with respect to the fixed date method (Huss et al., 2009), or to investigate the impact of changing glacier area and hypsometry to values of mean specific mass balance (Paul, 2010; Huss et al., 2012). Extensive accumulation measurements (e.g., Sold et al., 2016) provide detailed information on the intractable problem of spatial variability of snow accumulation while additional optical imagery provides information on the evolving snow cover (e.g., Huss et al., 2013; Barandun et al., 2015; Kronenberg et al., 2016).

Despite the relatively high number of existing studies related to the reanalysis of glacier mass balance records, few of them (e.g., Thibert et al., 2008; Eckert et al., 2011) include a rigorous uncertainty analysis. Consequently, error assessment and reanalysis of glaciological data is of central interest to both the glaciological and climatological community (Zemp et al., 2013, 2015).

In this paper we present a reanalysis of the mass balance record of a small alpine glacier including a thorough uncertainty assessment. The example glacier is a particularly useful case as, like for many other glacier mass balance records, the measurement network has changed over time and the data record suffers from inconsistencies that must be tackled in order to create a consistent homogenized time series. Thus, developing a reanalyzed record and providing a detailed analysis of the uncertainty associated with this record showcases a method that can be applied to glacier data-sets suffering from similar inconsistencies and provides insights into the reliability of existing glacier mass balance series for which such error analyses are no longer possible or practical. The reanalysis presented here involves:





- (i) The creation of a complete and consistent set of point mass balance by correcting for methodological shortcomings in the original data and by generating pseudo point measurements for years and locations without measurements applying a physically-based surface mass balance model.

- (ii) The re-calculation of glacier wide mass balance based on a variety of extrapolation techniques and the integration of new topographic data from ortho-images and airborne laser-scanning (ALS) in order to minimize the effect of changing glacier outlines on the glacier wide mass balance.

- (iii) A sound uncertainty assessment regarding the results of this study including a comparison of the reanalyzed cumulative mass balance to the mass change obtained from the geodetic method based on airborne laser-scanning data.

## 2 Study site and data

Langenferner (Vedretta Lunga) is a small valley glacier situated at the head of the Martell Valley (46.46° N, 10.61° E) in the Ortler-Cevedale Group, Autonomous Province of Bozen, Italy (Fig. 1). It covers an area of 1.61 km$^2$ (2013) and the highest point of the glacier is an ice divide at around 3370 m a.s.l. The terminus elevation is 2711 m a.s.l. and the median altitude is 3143 m a.s.l. (Galos et al., 2015). The upper glacier part is mainly exposed to the north while the lower sections face east. Ground penetrating radar measurements in spring 2010 gave a glacier volume of approximately 0.08 km$^3$, with a maximum thickness of more than 100 m in the upper glacier part. The glacier runoff feeds the river Plima which is a tributary to the river Etsch (Adige).

Langenferner is situated at the southern periphery of the inner alpine dry zone, thus the climate is shaped by relatively dry conditions due to precipitation shadow effects from surrounding mountain ranges. The largest part of annual precipitation is associated with air-flow from southern directions, often resulting from cyclonic activity over the Mediterranean, while the location south of the main Alpine divide means that fronts from the North are of only minor importance. Mean annual precipitation rates in the region range from about 500 kg m$^{-2}$ at the bottom of the Vinschgau (Valle Venosta) to 800 kg m$^{-2}$ at Zufritt Reservoir (1851 m a.s.l.), up to approximately 1200 kg m$^{-2}$ at Careser Reservoir Station, 2605 m a.s.l. (Carturan et al., 2012). During the study period the mean zero degree level as inferred from the AWSs used in this study was at an altitude of 2500 m a.s.l. As in most regions of the Eastern Alps, glaciers in the Ortler-Alps have been far from equilibrium during the past decade and have hence experienced drastic losses in mass, area and volume (e.g., Carturan et al., 2013a, b; D'Agata et al., 2014) and at least during the period 2005 to 2013 Langenferner was amongst the glaciers with the most negative mean specific mass balances of the Etsch-Catchment (Galos et al., 2015).

### 2.1 Glaciological measurements

Direct glaciological measurements at Langenferner were initiated by the University of Innsbruck on behalf of the Hydrological Office of the Autonomous Province of Bozen/Bolzano (HOB) in the hydrological year 2004. The program was established as a supplement to the mass balance program at Weißbrunnferner / Fontana Bianca which was considered (i) as potentially



threatened by rapid glacier retreat and (ii) deemed to be not representative for the region due to the specific setting of the glacier (Kaser et al., 1995). Since the start of the program, the initially provisional measurement network has gradually evolved (Fig. 2) and hence, has changed substantially over time, especially in terms of spatial stake distribution, which poses a particular challenge for understanding the spatio-temporal variability of the glacier mass balance.

In fall 2002, a number of ablation stakes was installed in the lower part of Langenferner and systematic readings began in October 2003, when additional stakes were drilled, still only covering the lover half of the glacier. In August 2005, the stake network was extended to the upper glacier sections by adding seven more stakes, the position of which was initially not accurately recorded. Systematic readings of the stakes in the upper glacier part at the end of the hydrological year were not performed regularly until the year 2009, when the measurement network was further refined by adding five additional stakes

to the upper glacier sections. In the course of the study period four stakes in the lower most glacier part were removed due to drastic glacier retreat. The position of stake 13 in the middle part of the glacier had to be changed in 2011 due to outcropping rock at the original location. The current stake network (October 2016) is made up of 28 ablation stakes which consist of elastic white PVC-tubes of roughly 2 m length, connected by a piece of rubber-hose. Drilling depths vary between 4 m in the uppermost glacier part and 12 m in the lower sections. Stake readings are performed once to six times a year, depending

on the local conditions at the individual stake. During the study period extensive accumulation measurements at the end of summer were only necessary in the years 2010 and 2013 since net accumulation was restricted to about 10 % of the glacier area in the other observation years. Up to the year 2007 no accumulation measurements at the end of the hydrological year were performed and from 2008 to 2013, accumulation was measured by means of one or two snow pits at more or less arbitrarily chosen locations and, if necessary, by a varying number of snow depth probings.

Winter balance measurements have been carried out annually in the first half of May performing numerous snow depth probings and four (or three) snow pits distributed over the glacier surface in each of which the bulk density of the snow pack was measured gravimetrically. While the number and location of the snow depth probings was not fixed throughout the observation period, the number and positions of the pits were more or less kept constant, except for the year 2009 when the large amount of winter accumulation resulted in omitting pit 3. Since the year 2013, the observational set-up includes two

automatic weather stations (AWSs) on and near the glacier and in spring 2014 a run-off gauge was installed 3 km down-stream of the glacier terminus. Seasonal and annual mass balances are regularly submitted to the World Glacier Monitoring Service (WGMS).

## 2.2 Meteorological data

The mass balance model used in this study requires meteorological data as input. These data originate from AWSs (1851 to

3325 m a.s.l.) in the vicinity of the glacier (Fig. 1) and were provided by the HOB. Hourly values of air temperature, relative humidity, and global radiation were taken from the station Sulden Madritsch, 2825 m a.s.l., located in an alpine rock cirque some 2.5 km north of Langenferner. The other three required meteorological input variables are wind speed, precipitation and atmospheric air pressure. Those data were not available at Sulden Madritsch for the entire study period. Consequently wind speed data was taken from the station at Schöntaufspitze, 3325 m a.s.l., 5 km north of Langenferner. Air pressure was down-





scaled from ERA-Interim reanalysis data of the nearest surface grid point (47°N|11°E). Daily precipitation sums originate from the station at the dam of Zufritt Reservoir, 1851 m a.s.l. in the Martell Valley, approximately 11 km northeast of the glacier (Fig. 1).

Since 2013 the Institute of Atmospheric and Cryospheric Sciences, University of Innsbruck (ACINN) operates two AWSs at Langenferner. One station is drilled into the ice of the upper glacier part at an altitude of 3238 m a.s.l. and is designed to measure all meteorological parameters needed to calculate the glacial surface energy balance. The second station was installed on solid rock ground close to the middle part of the glacier serving as a robust back up to bridge possible data gaps of the ice station. Data of those two stations were used to derive spatial gradients and transfer functions of meteorological data, as well as to optimize the radiation scheme of the applied mass balance model.

## 2.3 Digital terrain models from airborne laser-scanning

Data from three ALS-campaigns were used for this study. Respective surveys were conducted around September $14^{th}$, 2005, on October $4^{th}$, 2011 and on September $22^{nd}$, 2013. For the area around Langenferner the point density of the 2005 and 2013 data sets is 1.06 and 2.65 points per $m^2$ respectively (Galos et al., 2015) and the density of the 2011 data set is 2.84 points per $m^2$. High resolution (1m) digital terrain models (DTMs) of the study area were calculated from the original ALS point data for all three data sets, where the mean value of all points lying in a raster cell was used as the elevation value for the cell and cells without measurements were interpolated from surrounding grid cells.

## 2.4 Glacier outlines from ortho-images, ALS and GPS

Orthophotos from four acquisitions were used for updating the glacier area of Langenferner. Orthophotos for the years 2003, 2006 and 2008 were provided by the Autonomous Province of Bozen / Südtirol, while data for the year 2012 were available as a base-map within Esri-ArcGIS. The delineation of glacier area was done manually, which, for this small and well-known glacier, ensures the maximum accuracy. Outlines for the years 2005, 2011 and 2013 were derived from ALS data using high resolution hill-shades and DEM-differencing (Abermann et al., 2010). ALS data also provided valuable information for the delineation of debris-covered glacier margins as areas which had undergone no change in surface elevation between two acquisition dates could be classified as ice-free while debris-covered ice was still subject to some lowering (Abermann et al., 2010). The glacier outline of 2010 was assessed from extensive differential GPS measurements in early October 2010.

In the uppermost part of the glacier the outlines are confined by ice divides which were derived applying a watershed algorithm to the ALS-DEM 2005. Although the glacier surface topography in this area changed during the observation period, the impacts on the glacier flow direction are negligible and the outlines of the uppermost glacier part were consequently kept constant throughout the study period.





## 2.5 Snow line from terrestrial photographs and satellite images

Information on snow cover extent was used to tune the mass balance model for individual points on the glacier. In order to map the evolution of the transient snowline at Langenferner during the ablation period, we made recourse of an extensive set of terrestrial and aerial images mainly recorded during the field campaigns on the glacier. We used photos from more than 70 field work campaigns, private visits and dated photographs from the internet. A small number of Landsat scenes from different dates provided additional information on the snow cover extent on the glacier. These data were used to manually determine the approximate date when the snow of the previous winter had melted entirely at each given stake location. In most cases the date of snow melt could be determined with an estimated accuracy of ± five days, and in many cases probably even better.

## 3 Homogenization of data and methods

### 3.1 Point measurements

Besides a sparse and unevenly distributed measurement set-up during the first half of the study period affecting annual mass balances, the original record of winter balances was influenced by methodological inhomogeneities concerning the attribution of snow accumulation and ice ablation to the correct reference year or season. These problems have to be addressed in order to create a consistent and comparable record of annual and seasonal mass balance according to the fixed date system (e.g., Cogley et al., 2011).

#### 3.1.1 Stratigraphic correction of snow measurements

Accumulation of snow or firn is measured by means of snow probings and snow pits. Both techniques record the entire snow pack down to a characteristic reference layer which is typically given by the ice surface at the end of the previous ablation season or the firn surface at the date of the last local mass minimum. Hence, there is a need to correct the snow depth measurements in spring for snow fallen during the previous hydrological year (i.e. before October, $1^{st}$) in order to obtain values corresponding to the fixed date period. As part of the re-analysis process we accounted for this problem which was not considered in the original analyses up to the year 2008.

#### 3.1.2 Ice ablation in the hydrological winter period

While incorrect attribution of summer snow can affect both annual and seasonal mass balance, ice ablation in late autumn (after September $30^{th}$) only affects the seasonal mass balances. In most years of the study period ice ablation during late autumn at Langenferner was negligible since low elevated areas at the tongue are relatively small and receive only little insolation during that time of the year. Nevertheless, in October / November of the years 2004 and 2006 considerable melt took place in the middle and lower parts of Langenferner. While in the original analyses this issue was not considered, although stake data from late autumn field work were available, the point winter balances were corrected respectively during the reanalysis process.





### 3.1.3 Fixed date versus floating date

Measurements for the annual mass balance were always carried out very close to end of the hydrological year and, if necessary, stratigraphic corrections for snow cover were applied in order to meet the fixed date criteria. Original winter mass balances on the contrary were calculated following the floating date approach meaning that the water equivalent of the snow pack accumu-
lated since the preceding summer was measured during a field campaign in the first half of May and no further corrections were applied. In order to make the results of the seasonal balances comparable, we calculated the fixed date winter mass balance by scaling the measured and corrected point values of winter balance in order to obtain the water equivalent of snow at the end of the hydrological winter season (April $30^{th}$). This was done based on precipitation measurements at Zufritt Reservoir and on the ratio of accumulated precipitation during the measurement period (floating date) and during the hydrological winter season
(fixed date) as follows:

$$b_{fix} = b_{fld} \cdot \frac{\sum P_{fix}}{\sum P_{fld}}, \tag{1}$$

where $b_{fix}$ and $b_{fld}$ are the fixed date and floating date point values of winter balance and $\sum P_{fix}$ and $\sum P_{fld}$ are the precipitation sums recorded at Zufritt Reservoir during the fixed date and the floating date period respectively.

### 3.2  Point mass balance modeling

A major shortcoming of the original mass balance record at Langenferner is the lack of observation points in the upper glacier part during the first years of the study period. This affects the calculations of annual mass balance in the early observation years and the temporal consistency of the record. Therefore, the central aim of this study is to create a spatially and temporally consistent set of point annual mass balance. Due to the relatively short and inconsistent set of existing measurements, existing statistical approaches (e.g., Lliboutry, 1974; Thibert and Vincent, 2009; Eckert et al., 2011) are inapplicable. Instead we gen-
erate artificial measurement points in the poorly-represented upper glacier section by applying a physically based energy and mass balance model (Mölg et al., 2012). The model was run in its point configuration as the purpose of this study is not to use a model for the spatial extrapolation of point measurements on the glacier wide scale as done in a number of other studies (Huss et al., 2013; Barandun et al., 2015; Kronenberg et al., 2016; Sold et al., 2016), but is to reproduce a best possible estimate of annual balance values for point locations where ablation stakes were placed in the subsequent years. In this configuration, the
model performance can be validated directly using data from available stake measurements. A spatially distributed model set up would introduce larger errors regarding the mass balance at selected points, while the point model allows for a spatially flexible model tuning and strongly reduces errors due to shortcomings in the spatial extrapolation of meteorological variables (e.g., Carturan et al., 2015; Sauter and Galos, 2016; Shaw et al., 2016) and the choice of the optimal parameter setting (e.g., MacDougall and Flowers, 2011; Gurgiser et al., 2013). The application of a relatively complex physical model is justified by
the dominant influence of micro-meteorological variability, local topographic factors and the resultant large spatial variability of the surface mass balance, which can only be resolved in a sufficient way by a process based model. Furthermore, we aim at





providing a set of homogenized point mass balances instead of glacier wide balances only, since point balances have proven to be valuable information sources for investigations on glacier-climate interactions (e.g., Huss and Bauder, 2009).

### 3.2.1 Transfer of meteorological variables

Meteorological variables (Section 2.2) were extrapolated to the glacier using spatial transfer functions. Those functions were obtained using data from the on-glacier weather station in the upper part of the glacier for the period July 2013 to August 2015, when measurements at all AWS were available. Air temperature from Sulden Madritsch was extrapolated to the glacier applying an altitudinal lapse rate of $6.5 \times 10^{-3} K/m$ and an offset of -0.56 K reflecting the different microclimates over rock (Sulden Madritsch) and the on-glacier station. Both values were derived from a linear regression between measurements at Madritsch and the glacier station during the summers 2013 and 2014. Relative humidity was corrected for saturation over ice for below-freezing temperatures but the data was not further modified since no clear spatial pattern was detectable in the analysed data sets. Global radiation was used to calculate a cloud factor (e.g., Mölg et al., 2009b; Haberkorn et al., 2015) which was assumed to be spatially constant over the study area. This factor was used to drive the radiation scheme of the mass balance model which was optimized for the glacier using short- and long wave radiation data from the glacier. Wind speed from Schöntaufspitze was linearly scaled to match the observed wind speeds at the glacier station using a scaling factor of 0.67. ERA-Interim atmospheric air pressure was reduced to the altitude of the stake locations by the barometric equation, and since the mass balance model is relatively insensitive to small changes in air pressure, the temporal resolution of one hour was achieved by linear interpolation of the six hourly reanalysis data. Daily precipitation sums measured at Zufritt Reservoir (1851 m a.s.l., 11 km from the glacier) were used to assess hourly precipitation at the glacier, whereby the daily sums were temporally redistributed according to the course of relative humidity during the measurement day. Precipitation was only assigned to time steps when humidity exceeded the threshold of 93 % and the amount for a single time step was then scaled according to the magnitude of exceedance. If this threshold was not reached throughout the day but precipitation was measured, the threshold was lowered by steps of 5 %. This procedure was found to have a remarkable positive impact on the model performance. The sensitivity of modeled point mass balance to this correction can easily reach about $\pm 100 \, \mathrm{kg \, m^{-2}}$ on the annual scale compared to a model driven with daily precipitation sums equally distributed over 24 hours.

### 3.2.2 Monte Carlo optimization of model parameters

The mass balance model approach was set up as follows (Fig. 3): first the model was pre-calibrated applying realistic values for the model parameters which were either taken from the literature or from direct meteorological observations in the study area. The first guess model precipitation $P_{0_{model}}$ was generated applying a precipitation scaling factor $\Gamma_0$ to the measured and temporally re-distributed record of Zufritt $P_{obs.red.}$ in order to fit the model to the observed values of winter mass balance (Equation 2).

$$P_{0_{model}} = P_{obs.red} \cdot \Gamma_0, \tag{2}$$





This pre-calibration was done for the location of stake 22 (Fig. 4), a stake situated in the upper glacier part, near the center of the area where the point modeling was carried out. This stake was chosen as the relatively homogeneous terrain surrounding it makes it representative for a wider region of the glacier and it offers by far the highest number of stake readings in the upper region of Langenferner. It is hence the best choice for the optimization of model parameters, which was done applying

a Monte-Carlo approach (e.g., Machguth et al., 2008; Mölg et al., 2012) performing 1000 model runs with different parameter combinations in order to find the best model setting for the local conditions. The optimal parameter combination was then applied to all stake locations in the upper glacier. An individual Monte Carlo optimization for each stake was not possible due to the low number of available readings at some stakes.

### 3.2.3   Model tuning for individual stakes and years

Large uncertainties in process-based studies of glacier mass balance are commonly related to accumulation and its spatial distribution: altitudinal precipitation gradient, redistribution of snow due to wind and its large influence on spatial accumulation patterns, the temporal evolution of surface albedo and hence net radiation, etc. (e.g., Gurgiser et al., 2013; Machguth et al., 2008). In order to minimize uncertainties related to the unsatisfactory representation of local accumulation in the mass balance model, we made recourse of a calibration procedure which integrates available snow information. Therefor, the mass balance

model with the optimized parameter setting was tuned by replacing $\Gamma_0$ in equation 2 by the individual (for stake $i$ and year $a$) scaling factor $\Gamma_{i,a}$ which accounts for all site-specific properties related to accumulation. This tuning procedure was performed for every stake and year individually and in a way that the observed date of the emergence of the ice surface at the respective location was correctly reproduced by the model. An automated iterative approach ensured that the modeled date did not differ by more than one day from the observed date. Note that this approach is not applicable in years with a persisting snow cover

at the stake location. But during the first observation years (2004 to 2008) when measurements were partly missing, annual mass balances at Langenferner were very negative and accumulation at the end of the year was restricted to a few percent of the glacier area. For the few years and stakes with missing measurements and snow cover persisting throughout the ablation season, $\Gamma_{i,a}$ was derived based on linear regression with $\Gamma_i$-series of neighboring stakes. Values for $\Gamma_{i,a}$ vary in the range of 1.1 to 4.7 (Fig. 5), and curvature of the terrain and other wind related factors are clearly reflected in the spatial

$Gamma_{i,a}$-patterns, while inter-annual variability seems to be determined by meteorological phenomena, such as number and strength of storm events, dominant flow direction during the accumulation period or the absolute amount of accumulation since years with lower accumulation amounts tend towards larger $Gamma_{i,a}$-values.

### 3.3   Extrapolation from point to glacier scale

Five different methods were applied to extrapolate the reanalyzed values of annual and winter point mass balance to the glacier wide scale in order to obtain mean specific balances. We applied the traditional contour line method in two different ways and additionally made recourse of three purely objective methods in order to assess and investigate possible differences and uncertainties due to the applied analysis method. After the individual extrapolation of point measurements, all applied methods





calculate the total mass change $\Delta M$ by spatial integration of the specific mass change $b$ over the area $S$ based on the following equation:

$$\Delta M = \int_S b \cdot dS. \tag{3}$$

The mean specific mass balance $B$ is then calculated as follows:

$$B = \frac{\Delta M}{S}. \tag{4}$$

Equations 3 and 4 can be applied to the entire glacier area to obtain the mean specific glacier mass balance, or to each single 50 m altitude band in order to calculate the altitudinal mass balance profile and subsequently the equilibrium line altitude (ELA). The latter is calculated as the lower most intersection of the altitudinal mass balance profile with the $b = 0$ axis (Cogley et al., 2011). While annual ($B_a$) and winter ($B_w$) mass balances are based on measurements, summer mass balances are calculated

as a residual:

$$B_s = B_a - B_w. \tag{5}$$

For comparisons with the geodetic method there is the need for direct glaciological balances over the geodetic survey period. These are calculated summing up the annual glaciological mass changes ($\Delta M_a$) from the beginning ($t_0$) to the end ($t_1$) of the period of record ($PoR$) and dividing the result by the average glacier area $S$ through that period (equation 6).

$$B_{glac.PoR} = \frac{\sum_{t_0}^{t_1} \Delta M_a}{\frac{1}{2} \cdot (S_{t_0} + S_{t_1})}, \tag{6}$$

### 3.3.1 Contour Line based extrapolations

The contour line method (e.g., Østrem and Brugman, 1991) is an often used approach for the determination of glacier mass balance. It is based on manually derived lines of equal mass balance based on point measurements and has the advantage that the spatial pattern of surface mass change is relatively well reflected in the analysis if the method is applied thoroughly. The manual

generation of contour lines often incorporates the integration of further observational information such as the position of the snowline, date of ice emergence at individual locations, meteorological conditions on the glacier and other expert knowledge such as typical spatial patterns etc. This kind of information is difficult to capture in a purely objective or mathematical sense, nevertheless it often enhances the quality of the results and the spatial resolution of mass balance information.

Mass balance contour lines are then used to derive areas of equal mass balance where the mean value of the contour lines

is assigned to the area between the lines. However, for this study we applied the contour line method in two different ways: Once in its purely traditional form creating areas of equal mass balance between the lines of 250 $\mathrm{kg\,m^{-2}}$ equidistance, and once applying the Esri-ArcGIS interpolation tool *topo to raster*, which is based on the ANUDEM algorithm (e.g., Hutchinson, 2008; Hutchinson et al., 2011), to the hand drawn and digitized contour lines and the set of reanalyzed point values. The latter method results in mass balance rasters with a 1x1 m resolution which were subsequently spatially integrated to obtain the mean

specific mass balance (Equations 3 and 4).



### 3.3.2 Automatic extrapolations

In contrast to the contour line based analyses, automatic extrapolation methods avoid subjective influences, are fast and relatively simple to apply but are subject to restrictions in realistically reproducing the spatial distribution of surface mass balance. We apply three fully automatic extrapolation procedures: (i) the profile method based on a linear regression of point mea-
surements with altitude and the area-altitude distribution of the glacier (e.g., Escher-Vetter et al., 2009), (ii) the automatic extrapolation applying the *topo to raster* function and (iii) an inverse distance weighting procedure. While the contour based methods were applied making recourse of available information for the respective year, the three automatic methods were based on a reduced but temporally consistent set of reanalyzed point measurements which in this context means that the number and position of the measurement points used in the calculations was kept (almost) constant. This was done in order to avoid noise
related to changes in the measurement set up affecting the temporal mass balance signal. For winter balances the creation of a consistent set of point values was not possible due to large year-to-year differences in amount and spatial distribution of measurements.

### 3.4 Geodetic mass balance calculations

The geodetic mass balance of Langenferner for the period 2005 to 2013 and the sub-periods 2005 to 2011 and 2011 to 2013
was calculated based on differencing high resolution (1m) DEMs from ALS data (e.g., Abermann et al., 2010) of the respective years. The total volume change $\Delta V$ was calculated by integrating the elevation change $\Delta h$ at the individual pixel $k$ of length $r$ of the co-registered DEMs as follows (Zemp et al., 2013):

$$\Delta V = r^2 \cdot \sum_{k=1}^{K} \cdot \Delta h_k. \tag{7}$$

Subsequently the derived volume change is converted to a geodetic mass balance over the period of record following equation
20 8:

$$B_{geod.PoR} = \frac{\Delta M_{geod}}{\overline{S}} = \frac{\Delta V \cdot \overline{\rho}}{\frac{1}{2} \cdot (S_{t_0} + S_{t_1})}, \tag{8}$$

where $\overline{\rho}$ denotes a mean glacier density of $850\pm60\,\mathrm{kg\,m^{-3}}$ as proposed by Huss (2013) and $\overline{S}$ is the mean glacier area at the two acquisition dates calculated as the mean of the extents at the beginning and the end ($S_{t_0}$ and $S_{t_1}$) of the $PoR$.

### 3.4.1 Corrections for snow cover and survey date

The results of the geodetic surveys were corrected for differences in snow cover between the acquisition dates as follows:

$$\Delta M_{geod.corr} = \Delta M_{geod} + \left(\overline{h}_{s_{t_0}} \cdot \overline{\rho} - \overline{h}_{s_{t_0}} \cdot \overline{\rho}_{s_{t_0}}\right) - \left(\overline{h}_{s_{t_1}} \cdot \overline{\rho} - \overline{h}_{s_{t_1}} \cdot \overline{\rho}_{s_{t_1}}\right), \tag{9}$$

where $\Delta M_{geod.corr}$ denotes the geodetically derived mass change corrected for snow cover differences between the two acquisition dates $t_0$ and $t_1$, $\Delta M_{geod}$ is the uncorrected mass change, $\overline{h}_s$ denotes the mean snow depth (at dates $t_0$ and $t_1$ respectively), $\overline{\rho}$ the bulk glacier density and $\overline{\rho}_s$ the mean snow density at the acquisition dates $t_0$ and $t_1$.





In 2005, no field measurements were performed close to the ALS survey date. But field data from September $4^{th}$, 2005 in combination with meteorological records of nearby AWS, as well as photographs from nearby glaciers, indicate that there was basically no snow cover at the date of the 2005 ALS campaign. Seasonal snow was hence regarded as negligible for glacier wide analyses in 2005. In 2011, in situ measurements were performed on September $30^{th}$, four days prior to the ALS campaign. Despite the short time difference between direct and geodetic measurements, we applied a correction of the measured snow depths due to relatively warm weather conditions in this period. Therefor, the (optimized but untuned) mass balance model was initialized at all ablation stakes and a few additional locations using the measured snow depths and densities of September $30^{th}$ as initial condition. In 2013, extensive direct measurements were carried out simultaneously to the ALS campaign on September $23^{rd}$ in order to quantify the high amount of snow and firn in this year.

Survey date corrections were based on modeled mass change during the periods between ALS survey and direct measurements in the years 2005 and 2011, while in 2013 the correction was based on direct measurements performed on September $23^{rd}$ and September $30^{th}$. Point values of snow and mass change were extrapolated using *topo to raster* in order to calculate glacier wide mean values for the individual properties. Mean specific mass changes (representing corrections for snow cover and survey date respectively) were finally added (subtracted) to (from) the geodetically derived mass change over the survey period.

### 3.5 Annual glacier topographies and outlines

Changes in glacier area and topography may have significant impacts on the mass balance of mountain glaciers through various feed-backs (e.g., Paul, 2010; Huss et al., 2012) and since respective data is used as input for glacier models, they constitute glaciological key information. Hence, there is a need to frequently update topographic reference data used in mass balance calculations (e.g., Zemp et al., 2013, 2015). Langenferner was subject to remarkable hypsometry changes during the study period (Fig. 4 and 6). While glacier outlines for the current study could be directly derived from ortho-photos or ALS data for all years except for 2004, 2007 and 2009, data on glacier topography is only available through the three ALS campaigns. In order to minimize the effect of outdated area and hypsometry we calculated annual glacier outlines and topographies by combining the available set of related data with the fields of reanalyzed annual surface mass balance. For that we consider the change in surface elevation $\Delta h$ at one location (pixel) $k$ of the glacier over the time period t as the result of the following terms:

$$\Delta h_{k,t} = \Delta h_{surf_{k,t}} + \Delta h_{dyn_{k,t}} + \Delta h_{basal_{k,t}}, \tag{10}$$

where $\Delta h_{surf_{k,t}}$ denotes the surface elevation change related to surface mass balance, $\Delta h_{dyn_{k,t}}$ represents the surface change due to glacier dynamics and $\Delta h_{basal_{k,t}}$ is the surface change related basal (and internal) processes. As the latter term is assumed to be relatively small on the glacier wide scale (e.g., Cuffey and Paterson, 2010), it is neglected. The rasters of spatially extrapolated surface mass balance for each year (section 3.3.1) and those referring to snow and date corrections (section 3.4.1) can be summed up for the time period between the two geodetic surveys in order to calculate the term $\Delta h_{surf_{k,t}}$. Consequently,





$\Delta h_{dyn_{k,t}}$ for the respective period can be calculated as follows:

$$\Delta h_{dyn_{k,t}} = \Delta h_{k,t} - \sum_{t_0}^{t_1} \Delta h_{surf_{k,t}} = (h_{k,t_1} - h_{k,t_0}) - \sum_{t_0}^{t_1} \Delta h_{surf_{k,t}}, \tag{11}$$

where $h_{k,t_0}$ and $h_{k,t_1}$ are the surface elevations at the pixel $k$ given by the DEMs taken at date $t_0$ and $t_1$ respectively and $\sum_{t_0}^{t_1} \Delta h_{surf_{k,t}}$ refers to the temporally integrated elevation change due to surface mass balance. Due to the absence of data on the temporal evolution of glacier flow velocity, we assume the rate of $\Delta h_{dyn_k}$ to be temporally constant during the observation period. This simplifies the calculation of the annual $\Delta h_{dyn_{k,a}}$ to:

$$\Delta h_{dyn_{k,a}} = \frac{\Delta h_{dyn_k}}{d_{PoR}}, \tag{12}$$

where $d_{PoR}$ is the duration of the observation period in years. The result is a raster of $\Delta h_{dyn_{k,a}}$ which can be applied to all the observation years. The surface elevation of a certain year $h_{k,a}$ can finally be calculated by adding the surface elevation change due to surface mass balance in the respective year $a$ and the annual change related to glacier dynamics to the surface elevation of the previous year $h_{k,a-1}$ (Equation 13).

$$h_{k,a} = h_{k,a-1} + \Delta h_{surf_{k,a}} + \Delta h_{dyn_{k,a}}. \tag{13}$$

The DEM taken at the end of the observation period serves as a boundary condition for surface elevation at areas which become ice-free during the observation period in a way that all raster cells in those areas showing a surface height smaller than the surface of the ice free topography are set back to the value of the latter. Glacier outlines were derived by identifying ice-free pixels as having undergone no change in surface elevation.

Note that the term $\Delta h_{dyn_{k,a}}$ represents all the differences between the direct surface measurements and geodetically detected surface changes. These are not only differences which can be associated which glacier dynamics, but also shortcomings in the spatial extrapolation of surface mass balance measurements and changes due to internal or basal processes. This problem does not affect the temporally-integrated topography change, but may lead to additional errors regarding annual surface topographies. Nevertheless, this simple method provides a possibility to annually update the reference area and topography used in the mass balance calculations and hence represents a useful tool for areas with large changes in surface elevation.

## 4  Uncertainty Assessment

In order to enhance the value of the reanalyzed mass balance record, a detailed error assessment was performed following the recommendations of Zemp et al. (2013). We categorized potential errors in the measurements and analyses into random ($\sigma$) and systematic ($\epsilon$) errors. In the subsequent sections we discuss the origin of such errors related to the methods applied and explain how they were assessed. Thereby we primarily focus on random uncertainties, since systematic errors are difficult to quantify in the absence of an absolute reference for validation. In order to detect an eventually significant systematic bias in the reanalyzed record of annual balance, we finally perform a comparison of directly measured mass changes to those obtained by the geodetic method. The individual errors sources and the respective numbers used in the uncertainty model are listed in the supplementary material of this paper.



### 4.1 Uncertainties of glaciological measurements

Uncertainties in glaciological mass balances may originate from various sources and can be categorized into errors in point measurements, errors related to spatial extrapolations of point measurements and errors due to inaccurate or outdated glacier extents (Zemp et al., 2013). The random error of the mean specific mass balance for an individual year ($\sigma_{glac.total.a}$) can

consequently be formulated as follows:

$$\sigma_{glac.total.a} = \sqrt{\sigma^2_{glac.point.a} + \sigma^2_{glac.spatial.a} + \sigma^2_{glac.ref.a}}, \qquad (14)$$

where $\sigma_{glac.point.a}$ is the error due to uncertainties on the point scale, $\sigma_{glac.spatial.a}$ represents errors related to spatial extrapolations and $\sigma_{glac.ref.a}$ accounts for uncertainties due to inaccurate glacier outlines.

#### 4.1.1 Uncertainties related to point measurements

Random errors on the point scale mainly originate from inaccurate readings. This involves ablation and accumulation measurements equally. For ablation stakes the respective error is in the order of two or three centimeters. Limited representativeness of an ablation stake due to surface roughness is not really an error on the point scale but can introduce errors to the analysis when the data is extrapolated to larger areas. At Langenferner such surface features are typically $\leq 20\,\mathrm{cm}$ in height, although after long periods of exceptionally strong melt, surface structures were observed to reach the order of 30 to 50 cm in the lowest

sections of the glacier.

For accumulation measurements the error potential is generally higher. Snow pits with measurements of snow depth and density offer the highest accuracy ($\approx 50\mathrm{kg\,m}^{-2}$). But the number of snow pits is often kept low, as they are labor intensive and time-consuming. Snow probings are somewhat less accurate since they are affected by instrument tilt, by uncertainties in the spatial extrapolation of snow density and by possible difficulties in the determination of last summer's reference surface. The

latter effect can lead to large errors on the point scale but is assumed to play a minor role in this study since most "outliers" could be identified due to the high number of probings in combination with snow pit information.

However, the impact of uncertainties in point measurements on glacier wide calculations depends on the amount of point measurements and their spatial distribution. To quantify this problem, we applied a bootstrap approach (e.g., Efron, 1979) in which random errors according to a defined normal distribution were applied to all available individual point measurements

before calculating the glacier wide balance 5.000 times for each case using the inverse distance weighting method for extrapolation. The respective annual uncertainties are then given by the standard deviation of the 5.000 runs and range from 11 to 26 $\mathrm{kg\,m}^{-2}$ for annual balances and from 7 to 16 $\mathrm{kg\,m}^{-2}$ for winter balances.

#### 4.1.2 Uncertainties in the extrapolation of point measurements

Similar to the above problem, uncertainties related to the applied extrapolation method are also dependent on the number and

distribution of point measurements, as well as to spatial balance patterns of the individual year or season. We assessed those uncertainties based on the analysis of the glacier wide reanalyzed mass balances obtained from the five extrapolation methods



used. The annual extrapolation uncertainty in our study is finally represented by the absolute range of the bias corrected results ($r_{Brea_{bc}}$ in Table 1) ranging from 24 $\mathrm{kg\,m^{-2}}$ (2013) to 134 $\mathrm{kg\,m^{-2}}$ (2008). Respective values for winter vary between 31 $\mathrm{kg\,m^{-2}}$ (2013) and 95 $\mathrm{kg\,m^{-2}}$ (2004). Note that for winter balances we used the range without bias correction ($r_{Brea}$) since the biases between the individual extrapolation methods were small and their origin could not be unequivocally explained.

### 4.1.3 Uncertainties due to inaccurate glacier outlines

Since for this study we make use of annual glacier outlines derived from orthophotos, ALS, or calculated as described in section 3.5, our analyses are not systematically affected by this issue. Hence the remaining uncertainties related to this problem are given by the random uncertainties of the annual glacier areas. We estimated the related standard error to be 15 $\mathrm{kg\,m^{-2}}$. For the year 2005 we applied a more conservative estimate of 25 $\mathrm{kg\,m^{-2}}$, since the reference area for this year suffers from larger uncertainties due to the fact that it was derived by manually up-dating the 2003 glacier extent with a few GPS points taken in 2004.

### 4.2 Uncertainties in geodetic mass balances

Uncertainties in the geodetic mass balance are mainly related to two problems: (i) errors in the used DEMs and (ii) uncertainties related to the conversion of the observed surface elevation changes to changes in mass. The over-all random error of the corrected geodetic mass balances can be expressed as:

$$\sigma_{geod.corr} = \sqrt{\sigma^2_{geod.total} + \sigma^2_{dc} + \sigma^2_{sc} + \sigma^2_{sd}}, \tag{15}$$

where $\sigma_{geod.total}$ refers to the remaining uncertainties related to geodetic measurements after all applied corrections such as co-registration etc., $\sigma_{dc}$ is the error related to density conversion, $\sigma_{sc}$ refers to the error due to snow cover and $\sigma_{sd}$ is the remaining error due to different survey dates compared to the glaciological method. Note that equation 15 differs from equation 18 in Zemp et al. (2013) in two points: Firstly we split the uncertainties related to bulk glacier density and those introduced by differences in snow cover between the two survey dates. This is done because the available set of data allows for a sound quantification of snow cover. Secondly, we do not include the impacts of basal and internal processes since they neither represent an error in geodetic mass balance calculations nor could they be quantified in a sufficient matter in the frame of the current study. Nevertheless we discuss those effects in section 5.6.

### 4.2.1 Uncertainties in ALS measurements

In order to minimize systematic errors in the geodetic analyses, a co-registration of the three ALS data sets was performed including the roofs of three mountain huts in the vicinity of Langenferner. Thereby no significant dependence of the errors to slope and aspect of the surface could be detected. The remaining uncertainty potential related to vertical errors after co-registration and due to spatial auto-correlation for the individual survey periods $\sigma_{geod.total}$ was estimated as 0.15 m. This value is based on tests over reference areas not involved in the co-registration and can be regarded as a quite solid upper threshold of possible vertical errors over the glacier area.



### 4.2.2 Uncertainties related to glacier density

In our study uncertainties related to unknown mean glacier density are reflected by the applied density range of $850\pm60\,\mathrm{kg\,m^{-3}}$ (Huss, 2013). Based on the knowledge about the study area, such as the typical size of the accumulation area and the absence of large crevasse zones, we estimate the real near surface glacier density in the absence of seasonal snow to be in the range of

850 to 880 $\mathrm{kg\,m^{-3}}$.

### 4.2.3 Uncertainties due to snow cover and survey date differences

Uncertainties due to differences in snow cover at the two acquisition dates are difficult to estimate but we assume that they are quite small after we applied respective corrections (section 3.4.1). Similar is true for uncertainties related to different acquisition dates between geodetic and glaciological surveys. Especially for the two longer periods (2005 to 2013 and 2005 to

2011), due to the drastic mass loss both errors are at least one order of magnitude smaller than the uncertainties related to the used bulk glacier density and are hence of minor importance. However, the respective errors for both problems were estimated as $100\,\mathrm{kg\,m^{-2}}$ for all (sub-) periods.

### 4.3 Method comparison

In order to check the reanalyzed record of annual mass balance for a significant systematic bias, we compare the results for

the period 2005 to 2013 to the mass change inferred using the geodetic method. Doing so, it has to be considered that the two methods are subject to generic differences since the glaciological method only captures (near) surface mass changes, while the geodetic approach also detects volume (and mass) changes due to internal and basal processes. Consequently, we avoid using the term "validation" for the methodological cross check. Especially since we omit the explicit consideration of the above mentioned processes due to the fact that respective estimates without related measurements are speculative. However, method

comparisons were performed for the period 2005 to 2013, as well as for the sub-periods 2005 to 2011 and 2011 to 2013.

After applying the corrections described in section 3.4.1, the reduced discrepancy $\delta$ (Zemp et al., 2013) between the two methods can be calculated as

$$\delta = \frac{\Delta PoR}{\sigma_{common.PoR}} = \frac{B_{glac.PoR} - B_{geod.PoR}}{\sqrt{\sigma^2_{glac.PoR} + \sigma^2_{geod.PoR}}}. \tag{16}$$

Agreement between the two methods can be assumed within the 95 % (90 %) confidence interval if $|\delta| < 1.96$ ($|\delta| < 1.64$).

See Zemp et al. (2013) for a detailed description of this test.

## 5 Results and Discussion

### 5.1 Modeled annual point mass balance

Overall, 80 values of annual point mass balances were calculated using the presented model approach. For 33 of those cases field measurements are available which allows for independent validation of the applied approach, yielding a root mean square





deviation (RMSD) of 128 $\mathrm{kg\,m^{-2}}$ and an $R^2$ of 0.96 between modeled and measured values (Fig. 7). The magnitude of this error is similar to the uncertainty of glaciological point measurements reported in the literature (e.g., Thibert et al., 2008; Huss et al., 2009; Carturan et al., 2012), and since no significant systematic errors such as biases for single stakes or years are detectable, the 47 newly-created point values of annual mass balance constitute a valuable basis for the reanalysis of the glacier

wide annual balance. Note that in years with persisting snow cover the model could not be applied in its current form. Hence for 2010 and 2013 the mass balance of only a few stakes could be modeled. This reduced the number of validation points but did not affect the reanalysis procedure, since for these years measurements at all stake locations are available.

## 5.2   Mean specific annual mass balance

Mean specific annual mass balances and their altitudinal distribution were calculated based on the homogenized set of measured

and modeled point values, applying five different extrapolation methods and using the set of newly created annual glacier outlines and topographies. The results for the two contour-line based extrapolation methods are almost identical and differ by only 0 to 5 $\mathrm{kg\,m^{-2}}$ ($RMSD < 2\,\mathrm{kg\,m^{-2}}$). Consequently, we chose the results obtained by the raster-based contour line method as our reference since this method has the advantage of being less labor intensive than the traditional contour method and it results in high resolution (1x1 m) grids of surface mass balance which were also used to calculate annual glacier topographies

and outlines (Section 3.5).

The results show a persistent mass loss in all observation years. For the reference method, single year numbers vary between -1556±47 $\mathrm{kg\,m^{-2}}$ in 2012 and -247±31 $\mathrm{kg\,m^{-2}}$ in 2013, with a study period average of -1138±80 $\mathrm{kg\,m^{-2}}$ (Fig. 8 and Table 1). While all applied extrapolation methods display a common signal in terms of inter-annual mass balance variability ($R^2 > 0.98$), the three automatic extrapolations yield mass-balances which are considerably more negative than those obtained

by the contour line approaches (Table 2). The respective biases are -249 $\mathrm{kg\,m^{-2}}$ for the *topo to raster*-based automatic method, -188 $\mathrm{kg\,m^{-2}}$ for the profile method and -246 $\mathrm{kg\,m^{-2}}$ for inverse distance weighting (Fig. 9). Those negative biases can be well explained by the under-representation of accumulation areas in the consistent set of point balances on which both methods are based. This problem is not reflected in the contour line based calculations since those benefit from snow-line observations, sporadic accumulation measurements and at least a rough knowledge about the amount of accumulation and its

spatial distribution. The availability of a few continuous accumulation measurements at fixed locations would strongly reduce the biases of objective extrapolations and would enable the calculation of the glacier wide mass balance based on a reduced number of point observations and simplified extrapolations. However, this would lead to a loss of information on the spatial pattern of surface mass balance which constitutes an important component of modern and high level glacier mass balance monitoring as it is an important source of information for studies on energy balance and other glacier surface processes.

A comparison of the reanalyzed mass balance series to the original record shows that the two records strongly differ in their inter-annual variability ($R^2 = 0.84$), though the bias of the original series (-58 $\mathrm{kg\,m^{-2}}$) is relatively small (Fig. 9). Differences for single years are highest for the years 2004 and 2008 when they reach 384 $\mathrm{kg\,m^{-2}}$ and 317 $\mathrm{kg\,m^{-2}}$ respectively. For six of the ten observation years, differences between the original record and the reanalyzed series exceed the uncertainty range of the reanalyzed reference values. These large differences can mainly be attributed to two causes: (i) the lack of measurements





in the upper glacier part during the first half of the study period and the hence insufficient representation of spatial mass balance patterns in the extrapolation of point measurements, and (ii) the usage of outdated glacier extents which in our case biases the calculated glacier wide annual balances towards more negative values. The latter problem at Langenferner leads to a negative bias of typically about $20 \, \text{kg m}^{-2}$ after only one year. After only a few years without up-dating the glacier outlines

this effect can reach the order of $100 \, \text{kg m}^{-2}$. This matter is particularly affecting the original mass balance of 2004. For this first observation year, the analyses were based on glacier outlines of 1996 (Fig. 2) since newer topographic data at this time were not available. The results of all applied extrapolation methods, as well as the original balance numbers and glaciological key-numbers are listed in Table 1.

### 5.3    Mean specific seasonal balance

In contrast to annual mass balances, no modeling was involved in the calculations of the winter mass balances. However, the same extrapolation methods as used for the calculation of annual balances were applied to derive glacier wide winter balances. Again the two contour line based approaches displayed very similar results. For winter mass balances the differences between the two methods are slightly larger than for the annual balances which can be explained by smaller spatial balance gradients and consequently a lower spatial density of contour lines. Nevertheless, the differences for single years do not exceed $12 \, \text{kg m}^{-2}$.

The mean fixed date winter balance for the study period is $929 \pm 52 \, \text{kg m}^{-2}$ with a maximum of $1267 \pm 37 \, \text{kg m}^{-2}$ in the wet accumulation period of 2009. The exceptionally dry and warm winter 2007 resulted in the lowest value of $558 \pm 46 \, \text{kg m}^{-2}$. Note that in this winter period, the lower most parts of the glacier displayed negative mass balance due to considerable ice melt in late autumn 2006. Except for the year 2011, all reanalyzed winter mass balances are less positive than the original values (bias of original record = $71 \, \text{kg m}^{-2}$). This can be explained by the fact that all applied corrections in our case generally lower

the mass balance value and more positive values can only be the result of differences in the spatial extrapolation of point values or the use of different glacier extents which both have little impact ($< 50 \, \text{kg m}^{-2}$) on the winter balance at Langenferner due to generally small spatial (altitudinal) gradients in winter mass balance at this specific glacier.

The correlation between the original and reanalyzed records of winter mass balance is larger ($R^2 = 0.90$) than for annual balances which can be explained by the fact that the same set of point measurements has been used for both series and that

differences in glacier wide values can mainly be attributed to the corrections applied to the original data set (section 3.1). The homogenization of original winter balance point measurements revealed that the re-calculation of point winter balances according to the fixed date system generally showed the largest impact of the applied corrections. For the year 2010, the effect of this correction reached $140 \, \text{kg m}^{-2}$ on the glacier wide scale (17 % of the winter net accumulation). Corrections for snow of the previous hydrological year were showing a smaller effect but are still in the order of up to $100 \, \text{kg m}^{-2}$ on the glacier

scale. For 2011, when the corresponding corrections were already applied in the original series, skipping the correction would change the mean specific winter balance by more than $200 \, \text{kg m}^{-2}$. The impact of ice ablation during the hydrological winter period was greatest in the years 2004 and 2006 when it reached the order of 30 and $55 \, \text{kg m}^{-2}$ respectively. On the point scale respective values reach the order of $300 \, \text{kg m}^{-2}$ in the lower most glacier part which, in this very dry and warm winter, resulted





in a negative fixed date winter balance at the lower part of the glacier tongue. For glaciers with large tongues reaching low elevations or with large sun-exposed area fractions, this issue may be of even higher relevance than at Langenferner.

Summer balances suffer from the largest uncertainties since they are calculated as a residual from annual and winter mass balances and are hence affected by the uncertainties in both series. Absolute values of summer mass balance range from -2488 $\pm$73 $\mathrm{kg\,m^{-2}}$ in 2012 to -1463$\pm$47 $\mathrm{kg\,m^{-2}}$ in 2013. Differences between original and reanalyzed summer mass balances exceed the uncertainties of the reanalyzed series in seven out of ten observation years reaching up to 445 $\mathrm{kg\,m^{-2}}$ in 2004. A comparison of the two series yields $R^2$ of 0.70 while between the individual reanalysis series $R^2$ is $\geq 0.97$.

## 5.4 Geodetic mass balance 2005-2013

The mean surface elevation change at Langenferner during the eight-year period 2005 to 2013 amounts to -10.35$\pm$0.21 m. Surface elevation changes in the lower most glacier part reach the order of -40 m while in the highest regions changes in the order of one to three meters are detectable (Fig. 4). Assuming a bulk glacier density of 850$\pm$60 $\mathrm{kg\,m^{-3}}$, this corresponds to an uncorrected geodetic mass balance of -9397$\pm$687$\mathrm{kg\,m^{-2}}$. The correction for differences in snow cover between the two acquisition dates changes the result to -9596$\pm$694 $\mathrm{kg\,m^{-2}}$. Note that the values slightly differ from those presented by Galos et al. (2015) since the study in hand makes use of reanalyzed data sets. The results of the geodetic analyses (including those for the sub-periods 2005-11 and 2011-13) are summarized in Table 3.

## 5.5 Uncertainties in glaciological and geodetic balances

The largest source of uncertainties in the reanalized glaciological record is the spatial extrapolation of of point measurements. The largest spread between individual extrapolation methods is shown in the years 2008 and 2009 in which the negative off-sets of the automatic extrapolation methods are especially large. We attribute this to very strong spatial mass balance gradients in these two years given by the fact that mass balances at stake locations were quite negative, but at the same time snow of the previous winter could sustain throughout the summer in concavely shaped areas of the upper glacier part. While these patterns are reflected in the contour line based extrapolations, automatic methods did not capture this due to missing measurements in the respective areas. This shows the importance of the integration of accurate snowline observations in calculations of glacier mass balance. For winter balances the largest extrapolation uncertainties occur in in 2004 when only 22 point measurements are available. However, this number would be sufficient if the measurements were well distributed over the glacier area which was not the case in that year. For both, annual and winter mass balances, the second largest uncertainty source is given by the uncertainties of point measurements. For annual balances they are in the order of 25 $\mathrm{kg\,m^{-2}}$ while for winter balances they range from 7 to 16 $\mathrm{kg\,m^{-2}}$ due to the generally higher number of measurements combined with less distinct spatial mass balance gradients.

Uncertainties in the corrected geodetic balances are mostly determined by the applied density range of 850$\pm$60 $\mathrm{kg\,m^{-3}}$. Other error sources only account for a few percent of the total random error, except for the short period 2011 to 2013, when the remaining uncertainty of the DEMs exceeds the uncertainty related to the density assumption.





## 5.6 Glaciological versus Geodetic Method

Applying equation 16 to the results of our glaciological reference method yields $\delta$-values between 0.54 and 1 (Table 3) indicating that there is agreement between the glaciological and the geodetic results well within the 90 % confidence interval (Zemp et al., 2013; Sold et al., 2016). Hence, a further calibration of the reanalyzed glaciological record is not necessary.

5   The results of the profile method also fulfill the above criteria for all three (sub-) periods and could hence also be regarded as acceptable. The point to raster and inverse distance weighting methods fulfill the 90 % confidence criteria only for the period 2011 to 2013 but results are within the 95 % confidence bounds for the other periods (Table 4). However, the three automatic extrapolation methods yield results which are persistently more negative than the geodetic method (Fig. 10), while from a physical perspective, the geodetic method, especially during periods of strong glacier mass loss, can generally be expected to

10 display results more negative than the glaciological method due to the effect of internal and basal melt processes.

### 5.6.1 The role of basal and internal melt

Several studies have shown, that basal and internal melt can be important contributors to total glacier ablation, depending on the specific glacier and the climatic setting (e.g., Alexander et al., 2011; Oerlemans, 2013; Andreassen et al., 2016). Generally, the most important sources of energy for subsurface melt on temperate glaciers are related to the conversion of potential energy by

15 water run-off inside and at the base of the glacier. The water may originate from precipitation and other accumulation processes or may enter the glacier from outside. In the latter case the water may be warmer than $0°C$ and hence can offer an additional source of thermal energy. Other contributors to basal and internal melt are the geothermal heat flux and the conversion of potential energy related to glacier dynamics (deformation and basal friction).

   In order to provide a rough estimate of subsurface melt processes at Langenferner, we calculated the melt-contribution of

20 water run-off. We applied a similar approach than used by Andreassen et al. (2016), but instead of precipitation, we considered water released by melt (Thibert et al., 2008), which was approximated by the summer mass balance. Liquid precipitation instantly running off the glacier and water from outside the glacier were neglected since both play a minor role at Langenferner. For the period 2005 to 2013 our calculation gives a value of 178 $\mathrm{kg\,m^{-2}}$. Melt caused by geothermal heat and glacier dynamics is estimated based on values in the literature (e.g., Thibert et al., 2008; Alexander et al., 2011; Sold et al., 2016) as 10

25 $\mathrm{kg\,m^{-2}\,a^{-1}}$ and 1 $\mathrm{kg\,m^{-2}\,a^{-1}}$ respectively. However, combining the estimates for the individual contributors during the eight year period 2005 to 2013 results in a subsurface melt of 266 $\mathrm{kg\,m^{-2}}$, which explains about 37 % of the difference between the glaciological and the geodetic method during the same period.

   After re-calculating $\delta$ (Equation 16) taking the estimate for subsurface melt into account, the contour line based extrapolation methods show the best agreement with the geodetic results (Table 4). While the results of the profile method are still acceptable

30 on the 90 % confidence interval for the periods 2005 to 2013 and 2011 to 2013 and on the 95 % confidence interval for the period 2005 to 2011, the results of the other two automatic methods are not acceptable for 2005 to 2013 and 2005 to 2011 respectively.





## 6 Conclusions

In this paper we have presented a detailed work-flow for reanalyzing series of annual and seasonal glacier mass balances. The approach was applied to the ten-year record of Langenferner, a small glacier in the Italian Eastern Alps. Existing sets of annual and seasonal point mass balance data were homogenized based on methodological corrections and on pseudo observations of

point mass balance obtained by a physical model. Based on the homogenized point data, glacier wide mass balances were re-examined using a variety of extrapolation methods. Finally a detailed uncertainty assessment was performed including a cross check of glaciological results to those obtained by the geodetic method.

The reanalysis revealed that common problems often neglected in mass balance analyses can significantly disturb the derived interannual mass balance signal. Comparing the reanalyzed results to those of the original record yielded differences in annual

mean specific mass balances of up to 384 $\mathrm{kg\,m^{-2}}$. This by far exceeds the uncertainties of the renalyzed values, which are in the range of 31 to 136 $\mathrm{kg\,m^{-2}}$. Considering that two mass balance series for the same glacier and time period are compared, the correlation of the two records is rather low ($R^2 = 0.84$). This misfit for annual balances could mainly be attributed to missing point measurements for the upper glacier part in the original data series.

In the reanalysis, this drawback was overcome by applying a process based mass balance model. After a Monte-Carlo based

parameter optimization, the performance of the model was enhanced through individual precipitation tuning for every stake and year using the observed date of ice emergence as a constraint. The validation of modeled annual point balances against independent observations showed a RMSD of 128 $\mathrm{kg\,m^{-2}}$ which is comparable to the uncertainties of glaciological point measurements reported in the literature. The applied model approach can consequently be regarded as a useful tool to generate additional accurate point mass balance information and to integrate auxiliary information such as time lapse photos or satellite

images into glacier mass balance analyses.

Uncertainties due to missing updates of rapidly changing glacier geometries represent another important source of uncertainty for annual balances. In our case this problem causes errors in the order of 20 $\mathrm{kg\,m^{-2}}$ after only one year growing almost linearly within the study period. To tackle this problem we presented a method which enables the calculation of annual glacier outlines by combining geodetic information on glacier topography and measured surface mass balance.

For winter balances the correlation between original and reanalyzed record is higher ($R^2 = 0.90$) than for annual balances which can be explained by the generally sufficient amount and spatial distribution of winter mass balance measurements. Winter balances at Langenferner are also less sensitive to changes in the spatial distribution of measurements and to missing updates of glacier geometry since in most years there is no significant altitudinal gradient in winter mass balance. Differences between the original and reanalyzed series of winter mass balance mainly originate from the fixed date correction which was

applied in the course of the reanalysis. Corrections for snow from the previous hydrological year are also of considerable importance while ice melt at the beginning of the hydrological year only plays a role in two years.

A thorough uncertainty analysis revealed that the typical random uncertainty of the reanalyzed mass balances is in the order of 80 $\mathrm{kg\,m^{-2}}$ for annual and about 52 $\mathrm{kg\,m^{-2}}$ for winter mass balances. Numbers for single years/seasons range from 31 $\mathrm{kg\,m^{-2}}$ to 136 $\mathrm{kg\,m^{-2}}$. The largest part of the uncertainties can be attributed to the extrapolation of point values to the



glacier scale which apparently do not only depend on the amount and distribution of measurement points, but also on annual characteristics such as spatial balance gradients. The propagation of point scale uncertainties to the glacier scale constitutes the second largest error source in our study with typical values of $22\,\mathrm{kg\,m^{-2}}$ for annual and $10\,\mathrm{kg\,m^{-2}}$ for winter balances. Finally, the comparison of the cumulative reanalyzed glaciological mass balance over the period 2005 to 2013 to the geodetic mass

balance over the same period yields agreement between the two methods indicating that there is no significant bias between the two methods and a calibration of the glaciological results is hence not required.

While the bias correction of glaciological series based on geodetic measurements has become a common procedure in the reanalysis of glacier mass balance records, the current study also addresses the inter-annual mass balance variability, as well as related uncertainties, a problem which has yet rarely been considered. In order to increase the value of mass balance series

and to better understand underlying processes, future studies should address this matter by the integration of multi-source data combined with sound uncertainty analyses.

*Author contributions.* SG designed the study, conducted the gross part of the analyses and wrote the manuscript, CK processed ALS data sets and performed a series of GIS calculations, FM contributed to the study design and performed the boot-strap calculations, LN contributed to the paper design and writing, FC created most of the figures, LR provided the 2011 ALS data and information on ALS uncertainties,

WG performed the Monte-Carlo model optimization, TM provided the mass balance model and related information. GK helped to refine the manuscript and is the leader of the scientific project under which the study was carried out. All authors helped to improve the manuscript.

*Acknowledgements.* We thank all persons involved in the field work at Langenferner, with special thanks to Rainer Prinz. We are grateful to Roberto Dinale and Michaela Munari from the HOB for the constructive collaboration in coordinating the monitoring activities at Langenferner. Christoph Oberschmied provided his rich archive of photographs which was a great help in constraining the snow line evolution

at the glacier. The work of this study was financed by: Autonome Provinz Bozen – Südtirol, Abteilung Bildungsförderung, Universität und Forschung.



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





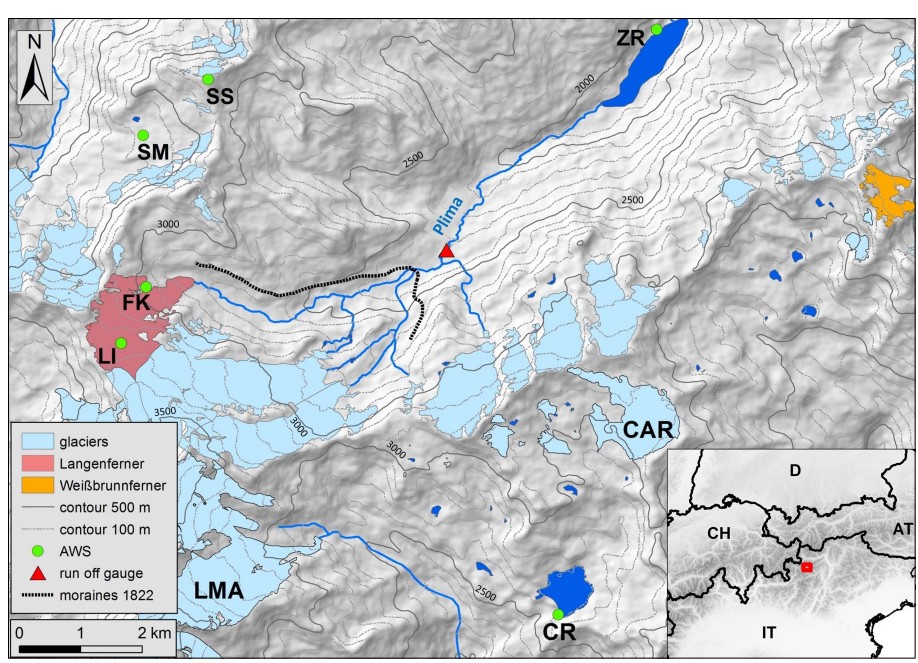

**Figure 1.** Overview of the study area. Langenferner is shown in red, Weißbrunnferner in orange, other glaciers in light blue. Green dots indicate AWSs referred to in this study: Zufritt Reservoir (ZR), Schöntaufspitze (SS), Sulden Madritsch (SM), Felsköpfl (FK), Langenferner Ice (LI) and Careser Dam Station (CR). The labels LMA and CAR refer to two other glaciers with mass balance measurements: La Mare and Careser (e.g., Carturan et al., 2012)



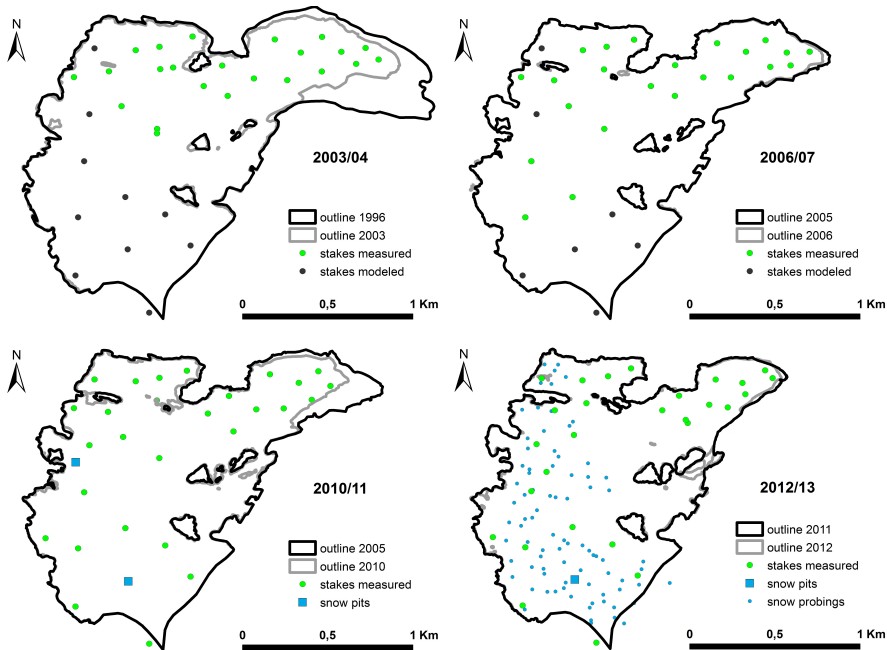

**Figure 2.** Changes in the annual data basis for mass balance calculations at Langenferner. Black glacier outlines are those used in the original analyses while gray lines refer to the reanalyzed outlines used in this study. Green and blue symbols indicate direct measurements which were used in the original analyses and after homogenization also in this study. Black dots symbolize modeled point values generated and used in the reanalysis.



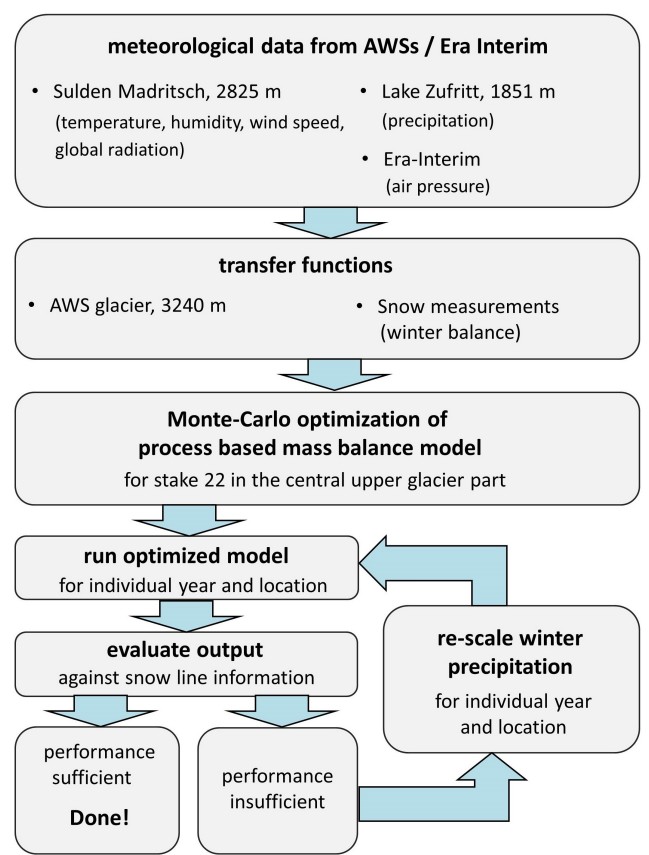

**Figure 3.** Flow chart of the applied model approach.





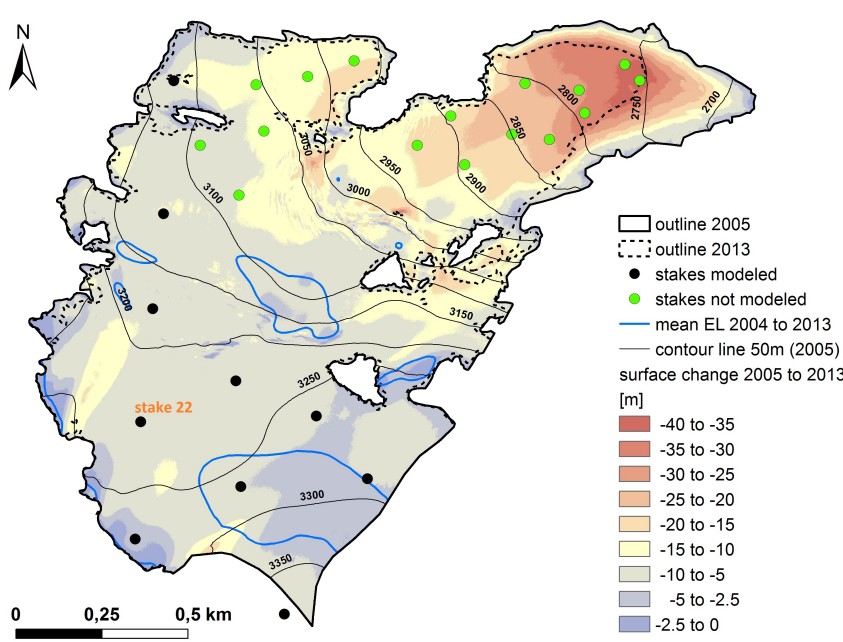

**Figure 4.** Surface elevation change derived from airborne laser-scanning for the period September 2005 to September 2013. Also shown are the locations of stakes used for the automatic extrapolation schemes, where black dots refer to stakes to which the mass balance model was applied. The blue line indicates the equilibrium line at the end of the hydrological years averaged over the study period.





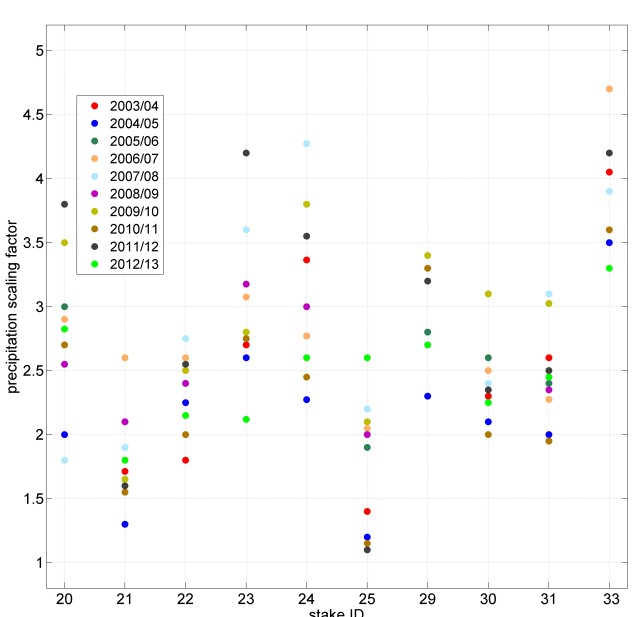

**Figure 5.** Precipitation scaling factors for different locations and years.





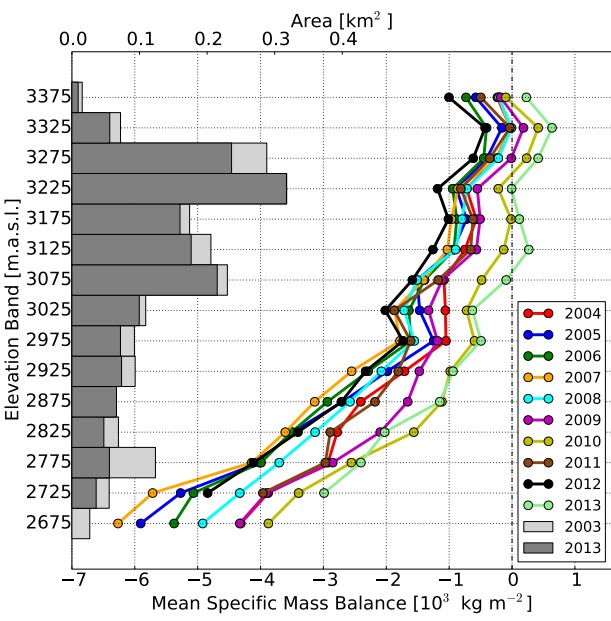

**Figure 6.** Vertical profiles of the reanalyzed annual mass balances and the altitudinal area distribution of the glacier at the beginning (2003) and the end of the study period (2012).





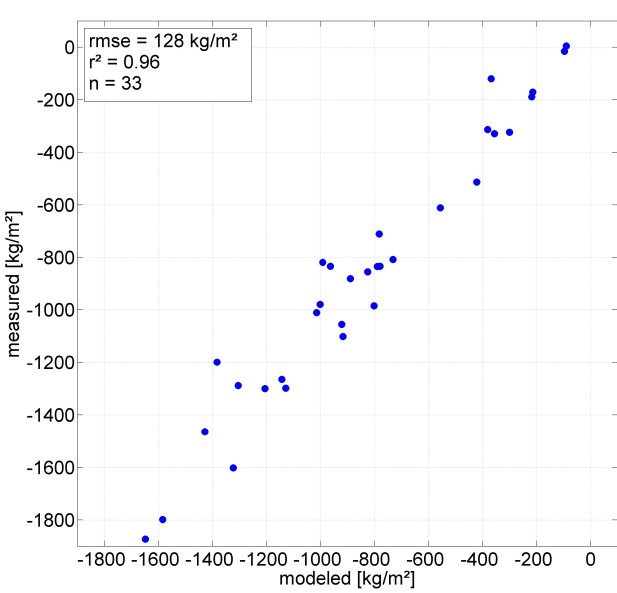

**Figure 7.** Scatter plot of modeled annual point mass balances against observations.





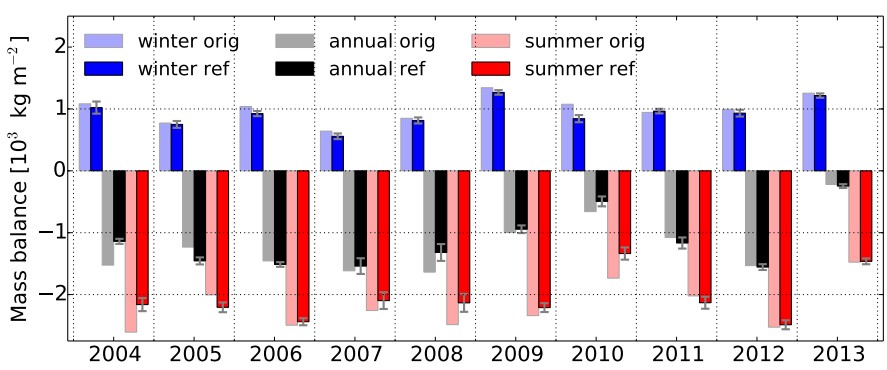

**Figure 8.** Original and reanalyzed seasonal mass balances at Langenferner during the study period.





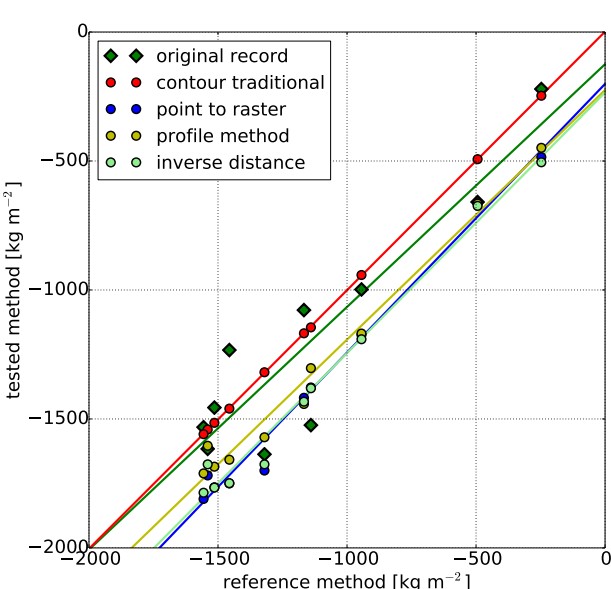

**Figure 9.** Scatter-plot of the annual results obtained by the different extrapolation methods against the reference values.





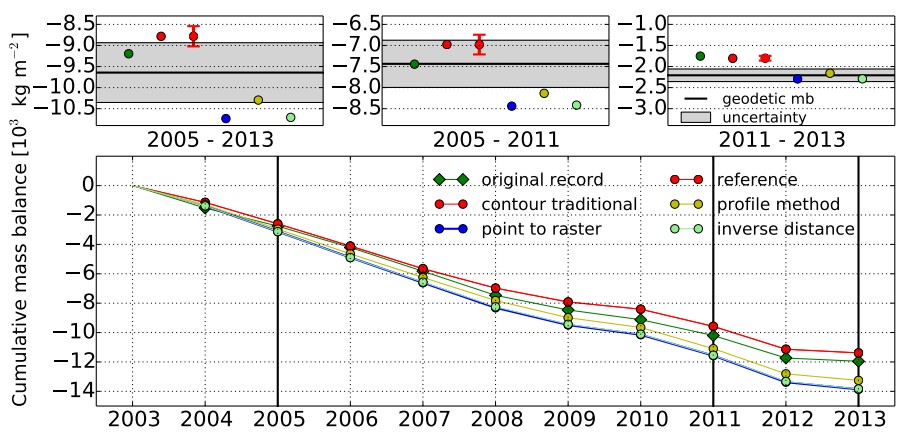

**Figure 10.** Comparison between geodetic and glaciological mass balances for three periods (upper three sub plots) and cumulative series of annual mass balance calculated using the five methods described in the paper.





**Table 1.** Reanalized and original annual and seasonal mass balances and glaciological key numbers for Langenferner. $B_{ref}$ refers to the reanalized mass balance calculated by the reference method, $\sigma_{glac.tot}$ is the related random uncertainty, $S_{rea}$ the glacier area at the beginning of the respective year, $ELA$ and $AAR$ are the corresponding equilibrium line altitude and accumulation area ratio, $B_{orig}$ refers to the original mass balance series. $B_{clt}$, $B_{ptr}$, $B_{prm}$ and $B_{ind}$ refer to the results of the other four extrapolation methods used (contour traditional, point to raster, profile method and inverse distance weighting. $_{orig-ref}$ is the difference between the original and the reference method, $r_{B_{rea}}$ the absolute range of the results of all five reanalysis methods and $r_{B_{rea.bc}}$ is the same range after a bias-correction of the individual series.

**Annual**

| Year | $B_{ref}$ | $\sigma_{glac.tot}$ | $S_{rea}$ | $ELA$ | $AAR$ | $B_{orig}$ | $B_{clt}$ | $B_{ptr}$ | $B_{prm}$ | $B_{ind}$ | $orig-ref$ | $r_{B_{rea}}$ | $r_{B_{rea.bc}}$ |
|---|---|---|---|---|---|---|---|---|---|---|---|---|---|
| | $\mathrm{kg\,m^{-2}}$ | $\mathrm{kg\,m^{-2}}$ | $\mathrm{km^2}$ | m a.s.l. | % | $\mathrm{kg\,m^{-2}}$ | $\mathrm{kg\,m^{-2}}$ | $\mathrm{kg\,m^{-2}}$ | $\mathrm{kg\,m^{-2}}$ | $\mathrm{kg\,m^{-2}}$ | $\mathrm{kg\,m^{-2}}$ | $\mathrm{kg\,m^{-2}}$ | $\mathrm{kg\,m^{-2}}$ |
| 2004 | -1140 | 41 | 1.938 | >3400 | 12 | -1524 | -1145 | -1378 | -1303 | -1381 | -384 | 241 | 28 |
| 2005 | -1456 | 59 | 1.864 | >3400 | 5 | -1233 | -1460 | -1750 | -1658 | -1749 | 223 | 294 | 47 |
| 2006 | -1514 | 38 | 1.833 | >3400 | 4 | -1456 | -1515 | -1765 | -1685 | -1766 | 58 | 252 | 23 |
| 2007 | -1540 | 127 | 1.821 | >3400 | 5 | -1616 | -1541 | -1719 | -1604 | -1676 | -76 | 179 | 124 |
| 2008 | -1320 | 136 | 1.785 | >3400 | 11 | -1637 | -1319 | -1700 | -1571 | -1676 | -317 | 381 | 134 |
| 2009 | -944 | 62 | 1.754 | 3278 | 18 | -998 | -942 | -1174 | -1169 | -1191 | -54 | 249 | 55 |
| 2010 | -494 | 82 | 1.722 | 3249 | 37 | -659 | -493 | -664 | -665 | -674 | -165 | 181 | 79 |
| 2011 | -1167 | 90 | 1.693 | >3400 | 7 | -1078 | -1168 | -1418 | -1442 | -1433 | 89 | 275 | 87 |
| 2012 | -1556 | 47 | 1.659 | >3400 | 1 | -1532 | -1559 | -1810 | -1711 | -1786 | 24 | 254 | 39 |
| 2013 | -247 | 31 | 1.620 | 3088 | 50 | -221 | -247 | -485 | -449 | -505 | 26 | 258 | 24 |

**Winter**

| Year | $B_{ref}$ | $\sigma_{glac.tot}$ | $S_{rea}$ | $ELA$ | $AAR$ | $B_{orig}$ | $B_{clt}$ | $B_{ptr}$ | $B_{prm}$ | $B_{ind}$ | $orig-ref$ | $r_{B_{rea}}$ | $r_{B_{rea.bc}}$ |
|---|---|---|---|---|---|---|---|---|---|---|---|---|---|
| 2004 | 1022 | 98 | 1.938 | <2600 | 100 | 1083 | 1021 | 976 | 927 | 961 | 61 | 95 | 68 |
| 2005 | 750 | 55 | 1.864 | <2600 | 100 | 772 | 754 | 716 | 706 | 711 | 22 | 48 | 20 |
| 2006 | 925 | 42 | 1.833 | <2600 | 100 | 1039 | 934 | 896 | 909 | 900 | 114 | 38 | 19 |
| 2007 | 558 | 46 | 1.821 | 2736 | 97 | 642 | 554 | 530 | 547 | 515 | 84 | 43 | 33 |
| 2008 | 814 | 49 | 1.785 | <2600 | 100 | 849 | 812 | 786 | 788 | 769 | 35 | 45 | 20 |
| 2009 | 1267 | 37 | 1.754 | <2600 | 100 | 1343 | 1270 | 1265 | 1296 | 1273 | 76 | 31 | 56 |
| 2010 | 843 | 58 | 1.722 | <2600 | 100 | 1076 | 839 | 820 | 787 | 795 | 233 | 56 | 29 |
| 2011 | 965 | 37 | 1.693 | <2600 | 100 | 944 | 967 | 938 | 950 | 970 | -21 | 33 | 39 |
| 2012 | 932 | 55 | 1.659 | <2600 | 100 | 995 | 944 | 892 | 910 | 903 | 63 | 52 | 29 |
| 2013 | 1216 | 35 | 1.620 | <2600 | 100 | 1255 | 1215 | 1214 | 1204 | 1234 | 39 | 31 | 47 |

**Summer**

| Year | $B_{ref}$ | $\sigma_{glac.tot}$ | $S_{rea}$ | $ELA$ | $AAR$ | $B_{orig}$ | $B_{clt}$ | $B_{ptr}$ | $B_{prm}$ | $B_{ind}$ | $orig-ref$ | $r_{B_{rea}}$ | $r_{B_{rea.bc}}$ |
|---|---|---|---|---|---|---|---|---|---|---|---|---|---|
| 2004 | -2162 | 106 | 1.938 | >3400 | 0 | -2607 | -2166 | -2354 | -2230 | -2342 | -445 | 192 | 94 |
| 2005 | -2206 | 79 | 1.864 | >3400 | 0 | -2005 | -2214 | -2466 | -2363 | -2460 | 201 | 260 | 38 |
| 2006 | -2439 | 57 | 1.833 | >3400 | 0 | -2495 | -2449 | -2661 | -2594 | -2666 | -56 | 227 | 13 |
| 2007 | -2098 | 135 | 1.821 | >3400 | 0 | -2258 | -2095 | -2249 | -2152 | -2191 | -160 | 154 | 126 |
| 2008 | -2134 | 145 | 1.785 | >3400 | 0 | -2486 | -2135 | -2486 | -2359 | -2445 | -352 | 355 | 131 |
| 2009 | -2211 | 72 | 1.754 | >3400 | 0 | -2341 | -2212 | -2439 | -2464 | -2464 | -130 | 253 | 94 |
| 2010 | -1337 | 99 | 1.722 | >3400 | 0 | -1735 | -1332 | -1484 | -1452 | -1469 | -398 | 152 | 88 |
| 2011 | -2132 | 97 | 1.693 | >3400 | 0 | -2022 | -1235 | -2401 | -2392 | -2403 | 110 | 271 | 99 |
| 2012 | -2488 | 73 | 1.659 | >3400 | 0 | -2527 | -2503 | -2702 | -2620 | -2689 | -39 | 214 | 41 |
| 2013 | -1463 | 47 | 1.620 | >3400 | 0 | -1476 | -1462 | -1699 | -1652 | -1739 | -13 | 277 | 61 |





**Table 2.** Statistical evaluation of mass balance series based on different extrapolation methods compared to the reference method. $Bias$, $R^2$, root mean square deviation before ($RMSD$) and after ($RMSD_{bc}$) a bias correction of the results.

| **Annual** | $B_{orig}$ | $B_{clt}$ | $B_{ptr}$ | $B_{prm}$ | $B_{ind}$ |
|---|---|---|---|---|---|
| $Bias$ | -58 | -1 | -249 | -188 | -246 |
| $R^2$ | 0.84 | 1.00 | 0.99 | 0.98 | 0.98 |
| $RMSD$ | 186 | 2 | 255 | 196 | 252 |
| $RMSD_{bc}$ | 177 | 2 | 56 | 56 | 56 |
| **Winter** | $B_{orig}$ | $B_{clt}$ | $B_{ptr}$ | $B_{prm}$ | $B_{ind}$ |
| $Bias$ | 71 | 2 | -21 | -27 | -26 |
| $R^2$ | 0.90 | 1.00 | 0.99 | 0.98 | 0.99 |
| $RMSD$ | 96 | 5 | 29 | 41 | 36 |
| $RMSD_{bc}$ | 65 | 5 | 19 | 31 | 25 |
| **Summer** | $B_{orig}$ | $B_{clt}$ | $B_{ptr}$ | $B_{prm}$ | $B_{ind}$ |
| $Bias$ | -128 | -3 | -227 | -161 | -220 |
| $R^2$ | 0.70 | 1.00 | 0.98 | 0.97 | 0.97 |
| $RMSD$ | 241 | 7 | 234 | 175 | 229 |
| $RMSD_{bc}$ | 204 | 6 | 57 | 68 | 65 |

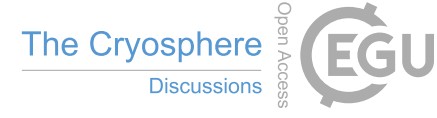

**Table 3.** Results of the geodetic analyses and the cross check between glaciological and geodetic method. Where $PoR$ stands for the observation period, $\Delta Z$ is the mean surface elevation change, $\Delta V$ the volume change, $B_{geod}$ is the uncorrected geodetic balance assuming a bulk density of 850 $\mathrm{kg\,m^{-3}}$, $corr_{sc}$ and $corr_{sd}$ refer to the corrections for snow cover and survey dates, $B_{geod.corr}$ refers to the corrected geodetic balance, $\sigma_{geod.total.PoR}$ is the total random error of $B_{geod.corr}$, $B_{glac.PoR}$ is the cumulated glaciological mass balance, $\sigma_{glac.total.PoR}$ is the corresponding random uncertainty, $\Delta_{rel}$ is the relative difference between glaciological and geodetic results and $\delta$ is the reduced discrepancy (Zemp et al., 2013).

| $PoR$ | $\Delta Z$ | $\Delta V$ | $B_{geod}$ | $corr_{sc}$ | $corr_{sd}$ | $B_{geod.corr}$ | $\sigma_{geod.total.PoR}$ | $B_{glac.PoR}$ | $\sigma_{glac.total.PoR}$ | $\Delta_{rel}$ | $\delta$ |
|---|---|---|---|---|---|---|---|---|---|---|---|
| | m | $10^6\,\mathrm{m^3}$ | $\mathrm{kg\,m^{-2}}$ | $\mathrm{kg\,m^{-2}}$ | $\mathrm{kg\,m^{-2}}$ | $\mathrm{kg\,m^{-2}}$ | $\mathrm{kg\,m^{-2}}$ | $\mathrm{kg\,m^{-2}}$ | $\mathrm{kg\,m^{-2}}$ | % | - |
| 2005-13 | -10.35 | -18.98 | -9397 | -198 | -49 | -9644 | 709 | -8971 | 240 | 7.5 | 0.90 |
| 2005-11 | -8.34 | -15.28 | -7439 | -38 | 41 | -7436 | 560 | -7111 | 234 | 4.4 | 0.54 |
| 2011-13 | -2.20 | -3.66 | -1908 | -169 | -8 | -2084 | 248 | -1830 | 56 | 12.2 | 1.00 |





**Table 4.** Reduced discrepancies $\delta$ for all extrapolation methods used in this reanalysis and for the original mass balance record. The upper panel shows results without the consideration of basal and internal melt, while the lower panel ($^*$) refers to $\delta$s calculated accounting for subsurface melt. Bold values refer to agreement on the 90 % confidence interval.

| $PoR$ | $\delta_{ref}$ | $\delta_{orig}$ | $\delta_{clt}$ | $\delta_{ptr}$ | $\delta_{prm}$ | $\delta_{ind}$ |
|---|---|---|---|---|---|---|
| 2005-13 | **0.90** | **0.32** | **0.90** | -1.74[1] | **-1.14** | -1.70[1] |
| 2005-11 | **0.54** | **-0.24** | **0.54** | -1.90[1] | **-1.38** | -1.86[1] |
| 2011-13 | **1.00** | **1.19** | **0.99** | **-0.95** | **-0.41** | **-0.92** |
| 2005-13$^*$ | **0.54** | **-0.04** | **0.54** | -2.10[2] | **-1.49** | -2.05[2] |
| 2005-11$^*$ | **0.20** | **-0.58** | **0.20** | -2.24[2] | -1.72[1] | -2.20[2] |
| 2011-13$^*$ | **0.75** | **0.95** | **0.74** | **-1.19** | **-0.65** | **-1.17** |

[1] Agreement on the 95 % confidence interval. [2] Not acceptable on the 95 % confidence interval.