# Peer review of "Reanalysis of a ten year record (2004-2013) of seasonal mass balances at Langenferner / Vedretta Lunga, Ortler-Alps, Italy"

_The Cryosphere, 2016_

## Referee Comment (RC2)

**Review:**

**Reanalysis of a ten year record (2004-2013) of seasonal mass balances at Langenferner / Vedretta Lunga, Ortler-Alps, Italy**

Stephan Peter Galos et al. The Cryosphere Discussions

**General comments**

The authors reanalyze a 10-year record of mass balance of a small glacier in the Italian Alps. They use previously applied approaches combined with new approaches. They present a thorough error calculation. The methods and results are in general carefully described. They make use of pseudo-observations to reanalyze the data and calculate the glacier wide mass balance. This study is thus not only a modelling experiment, but presented as a reanalysis of mass balance providing updated series of glacier mass balance. Such mixing of modelling and field-data is not unproblematic, but the authors do not discuss this. I miss a section in the paper discussing this choice. Furthermore, I miss a paragraph where the authors discuss the implementation of their results and flagging of the series. Information on number of pseudo stakes used for every year and which years that used modelled pseudo points in the calculation is lacking from the table. If some years need to be modelled and not others, this should be more clearly stated and flagged in the paper and resulting tables. A comment field in the table could be useful to so the series are marked with reanalysis and comment on modelling degree. Fig 2 could be extended with all 10 years so the data sources are shown.

**Data availability.** According to TC journal instructions, "Authors are required to provide a statement on how their underlying research data can be accessed". A section on Data availability therefore needs to be added to the paper with information on where to find the underlying data. The World Glacier Monitoring Service is the appropriate data service for much of the glaciological data.

**Specific comments**

Units. I suggest using m w.e. or m w.e. a-1 instead of kg/m2. Alternatively, mm w.e. Kg/m2 is not standard in the literature.

P2. L30. Very long sentence.

P3. Study site and data could be divided in 2 chapters.

P3. L27. Could mention the mean value for the period.

P4. 'The current stake network October 2016' is not that relevant here as the study period is up to and including 2013. Rather give the information for 2013. In 2.1. or in the introduction objectives (P3) the study period could be mentioned explicitly.

P4. 15. Add 'of remaining snow from the previous winter' or something similar to be clear to not mix up with winter balance/accumulation. Same for L18. Could mention how large area (in %) that had snow remaining in 2010 and 2013.

P4. L26. Mention the first mass balance year reported, e.g. 2003/2004, it may not be very clear since measurements began in 20002 (L5).

P5. L12. On 'Around', I assume also on the glacier since this is not specified. Replace around with covering or take out and just state the point density. Would it be better to resample to a 5 m model? The section 2.3 was quite brief.

P5. GPS should probably replace with GNSS (Global Navigation Satellite System) throughout the manuscript.

P5. L23. How much of the glacier is debris covered? This is not stated here or in chapter 2. A picture or aerial photo could have been a nice addition to the paper. Orthophotos are available according to the text. Could be added in an improved figure 1. The section and the referencing is a bit unclear to me, was the approach done following Abermann et al (2010) or was is it done by Abermann et al (2010).

P5. L28. Changes in ice divide is negligible. Could this be quantified?

P6. Ch 2.5. Could the extraction of snow line information be illustrated. Maybe add a table to the supplementary material? How were the imagery georeferenced and stakes identified on the imagery? Some more information could be added.

P6. L21. Write what you did instead of 'we accounted for this problem' to be specific.

P6. L28. This issue -> this melt …L29. Respectively -> for the additional melt

P7.L2-3 Unclear, do you mean that 'Whereas measurements were carried out close to the … and thus reported as fixed date in the original data, the….

P7. L10. When correcting accumulation: How are melting episodes accounted for?

P7. L17.  L 21. L23. Is -> was

P7. L32. Add that is also justified by available data to run the model (this is not always the case)

L29. Generated (use past tense on work carried out in this study)

P8. L 1. Provide to whom? 'Instead of  …only' -> 'in addition to glacier wide balances' One must be careful not mixing measured and modelled point data as this can add confusion, data should be clearly marked.

P9. L33. What does 'After the individual extrapolation of point measurements mean here? Interpolation could be a better word.

P10. L26. Mean manually drawn? Add this information.to be clear.

P10. L28. What does the latter method refer to here? Specify the method to be clear.

P11. Suggest to call this paragraph be called interpolation. Usually one refer to interpolation methods in general, not extrapolation methods. E.g., the Topo to Raster tool is an interpolation method

P11. L15. Why the e.g. reference here, unclear.as follows (Zemp et al., 2013) -> following Zemp et al. (2013).

P11. L19. Is-> was

P13. L13. Serves -> servedP14. L29. Substitute 'the above problem' with 'uncertainties in points'

P14. L10-12. Rough surface topography may give errors when measured differently by field observers. Uncertainties in identifying summer surface gives also uncertainties in point data.

P14. Line 29. Instead of starting a subchapter 'Similar to the above problem', state what the problem is.

P15. L10 due to the fact that it.. -> as it..

P15. 4.2.1. Could give some more details on the coregistration.

P17. L5. Specify why it could not be used.P20. L20. Than -> as

P18. Line 8. Are the original ELA and AAR values changed from the original record? Please make a comment addressing changes in ELA and AAR and add the original values to Table 1 if changed.

P19. L14. Add the value found by Galos et al. (2015).

P19. L15. Could add some results in the text for the short periods, not just referring to the table. Comment on the density conversion factor and uncertainty used for the short period, as short periods (1-3 yr) may have a different conversion factor. (Huss, 2013)

P20. L4. What do you mean with further calibration, do you consider some of the presented work as calibration?

P20. L20. Than -> as

P20. L21. Specify by adding 'reanalysed' summer balance values?

P20. L23. Add 'total', thus 'a total value

P20. L25. Specify what you mean with individual contributors.

P20. L28. Could add 'reduced discrepancy' after recalculating.

P20. L31. Name the two methods. What does it mean not acceptable.

P21. L1. Specify where you recommend it to be used.

P21. L4. Is it not the glacier wide averages that are calculated based on pseudo observations? Check L2-5.

P22. L8. As well as related -> of.  Inter-annual mass balance variability has indeed been considered before. Suggest rewriting.

Figure 1. Add source and year of glacier outlines. The map should be fitted to either one-column or two-column width. Could be extended in east to allow some more space around Weissbrunnferner. Borderline on inset could be refined and the inset would probably look better without the shaded background.

Figure 2. Same comment as 1, could be extend to 2-column width. The dense outer outline could be thinned.0,5 -> 0.5        Suggest to add all years here.

Figure 7. Add 1:1 lineFigure 8. Add (orig) and (ref) to the text.

Table 1. Add ELA and AAR if different from the original record. Add Area used in the original record. Add a comment field stating if modelled pseudo stakes are used for each year. Suggest having an additional table stating the data source available for each year including pseudo stakes used.

In general, there are numerous 'in order to' in the manuscript. 'in order to' can be replaced by 'to' in many (if not all) occurrences.

**Reference**
Huss, M.: Density assumptions for converting geodetic glacier volume change to mass change, The Cryosphere, 7, 877–887, doi:10.5194/tc-7-877-2013, 2013.

---

## Referee Comment (RC1) · E. THIBERT (Referee) · 8 Feb 2017

Review/comment on :

**Reanalysis of a ten year record (2004-2013) of seasonal mass balances at Langenferner / Vedretta Lunga, Ortler-Alps, Italy**

by S. P. Galos and co-authors,
submitted to The Cryosphere, tc-2016-286

E. Thibert, Grenoble (France) – 8 February 2017.

*General comment*
S. P. Galos and co-authors provide a data reanalysis scheme applied to a ten year record (2004 to 2013) of seasonal mass balance at Langenferner glacier in the European Alps.
The approach involves homogenization of available point values and reconstruction of missing data for years and locations without measurement by the application of a process-based model constrained by snow line observations. Point mass balances are then extrapolated to the overall glacier surface to quantify the glacier-wide balance help to different extrapolation techniques and recourse of topographic data. The 2 reconstructed seasonal series differ notably from the original records. The new annual mass balance series is compared to long-term volume changes determined from airborne laser-scanning data in a rigorous error analysis and following the framework proposed by Zemp and others (2013). The authors find good agreement between those determinations, the residual discrepancy being explainable by the natural scattering of the data. These favourable results are to confer a significant confidence in the correctness of the re-analysed glacier-wide mass balance times-series of Langenferner glacier.
The paper employs advanced methods of mass balance computations, and geo-statistical inferences. The paper is clear, well organized, and properly focuses the scope of the journal. I think this paper is to be welcomed also as a new appropriation from the glaciological community of the guidance proposed 4 years ago by Zemp and others (2013).

I have two main questions regarding the model tuning for point balances, and the spatial/temporal pattern of balances:

1°) I find the calibration of the accumulation rather weakly constrained on how the precipitation scaling factors $\Gamma_{i,a}$ are allowed to vary spatially (from stake-to-stake) and with time (from year-to-year). In the present formulation, $\Gamma$ is allowed to account for any deviation from the observed precipitations at all stakes and independently from what happens at other stakes. The authors mention on line 25 that the spatial pattern of $\Gamma_{i,a}$ is reasonable regarding terrain curvature and wind effects. Analysing Figure 5, it seems also that some stakes systematically have lower $\Gamma$ (stakes no. 21 or 25) and on some years all stakes deviates accordingly up (2009, 2011) or down (2010, 2004). Can the authors provide a colour coded map of the mean spatial structure of $\Gamma$ between 2004 and 2013 which could sustain this? Therefore one should expect a much more constrained formulation in Equation (2) when it comes to tune the model at individual stakes. Maybe the authors could reformulate $\Gamma$ as $\Gamma_{i,a} = \gamma_i \, \gamma_a$., fitting 2 uncorrelated functions holding spatial and times dependencies of accumulation, and accounting for systematic deviations at some locations ($\gamma_i$) and on some years ($\gamma_a$).

2°) Figure 6 shows that the altitudinal profiles of balance and the way they change over time somewhat follow the time-space decomposition proposed by Rasmussen (2004) or Kuhn (1984): b(z,t)=f(z)+δb(t).
In search of this, the authors could test if there is strong correlation between readings at the individual stakes. This feature also suggests that Lliboutry's model should work on the data set. A 10-year time series is just long enough and eminently suitable to test Lliboutry's linear model. I would therefore encourage to possibly test this analysis all the more that it could provide estimates for missing values of point balances in the accumulation area at the beginning of the record. Moreover these estimates would be free from any meteorological or glacier-to-climate assumption and safeguards the reconstructed series for an independent and unbiased analysis with climate drivers.

Here follow some detailed questions, comments, suggestions, and indications of minor typos in the paper. Please consider them in the same positive way in which they are proposed.

*Substantive comments*
P1-L1. You could mention that both uncertainties (scattering) and incorrectness (biases) affect calculations.

P1-L11. Annual values for mass balance refer to changes per unit of time (rate) and should here and throughout the manuscript expressed per year ($yr^{-1}$). Although there is not uniformity of units regarding specific mass balance (per unit of area), the authors could use the convenient, numerically equivalent and more meaningful mm w.e. or, alternatively, the SI unit m w.e. instead of $kgm^{-2}$.

P1-L19. Don't you think that a more direct (physically-based) driver is rather the surface energy balance?

P3-L21. Again please express rate units per year ($yr^{-1}$).

P4-L10. You mean that these stakes couldn't be measured any longer?

P4-L26. Please use when relevant "glacier-wide balance" terminology to fit Cogley et al. (2010) glossary.

P5-L24. A bit more needs to be say on how single out definitely areas outside the glacier subject to elevation loss (erosion in moraine terrain) and area on the glacier subject to reduce ablation (thick debris)?

P7-L19. A 10-year time series is just long enough to test Lliboutry's linear model as the spatial terms generally converge to steady values over this time scale.

P8-L15. At which spatial scale does ERA-Interim provides time series for atmospheric pressure?

P8-L26 to P9L8. How ranges the pre-calibrated scaling factor $\Gamma_0$ compared to the altitudinal gradient of precipitations?

P9-L10 to 28. As mentioned above, the calibration is very weakly constrained on how $\Gamma_{i,a}$ is allowed to vary spatially and with time and one should expect a much more constrained formulation in Equation (2). Please refer to my general comment ahead.

P10-L29. I suspect a 1x1m resolution grid to be an oversampled interpolation. Considering 250 kg m$^{-2}$ contour-lines and analysing Figures 4 and 6 suggests that just 4 to 5 contour-lines should cover the altitude range [3125-3375 m] over nearly 1 km in the accumulation area. Which means that a 100x100 m would be the right sampling scale adopting a maximum of 2 grid cells between contour-lines.

P11-L25. Don't you think it should be better to adjust field data to preserve the independency of the geodetic balance as a reference control?

P13-L15. Looks like my comment on P5-L24 on how single out areas outside the glacier subject to erosion and area on the glacier subject to low ablation.

P15-L5. Here and in section 4.2, I am not sure if 3 levels of subsections are worth.

P14-Equation 14. Here you assume errors to be uncorrelated to combine them quadraticaly. To some extent, this general formulation from Zemp et al. (2013) does not account that errors at a point and errors associated to the spatial integration are combined through weighting coefficients in the spatial averaging. Make here an explicit referral to the supplement S2.1 where details can be found about error calculations.

P16-L12. Unit kg m$^{-2}$ yr$^{-1}$?

P17-L5. How do you think your re-analysis model will perform in case of systematic positive balances over few years? Would you have to recalibrate it? Or to reformulated the approach?

P17-L8. Title section 5.2: Mean (specific) glacier-wide annual balance
Specific means per unit area, not averaged at the glacier scale.

P17-L9. Can you estimate the spatial variability of the mass balance at the glacier scale from the five different extrapolation methods? And for winter and summer balance as well?

P17-L20. Biases units kg m$^{-2}$ yr$^{-1}$?

P20-L10. It would be helpful to remind the reader that you did not correct for such internal processes.

*Tables*
Table 4. Legend. Remove "s" after $\delta$.

*Figures*
Figure 6. I don't find where the main text refers to Figure 6

Figure 7. I suggest to add the 1:1 line and the fitted line with the equation terms providing intercept (bias) and slope.

Figure 10. Is it possible to plot error bars (one sigma) on cumulative balances.

*References*
Cogley, J. G., Hock, R., Rasmussen, L. A., Arendt, A. A., Bauder, A., Braithwaite, R. J., Jansson, P., Kaser, G., Möller, M., Nicholson, L., and Zemp, M.: Glossary of Glacier Mass Balance and Related Terms, IHP-VII Technical Documents in Hydrology No. 86, IACS Contribution No. 2, UNESCO-IHP, Paris, 2010.

Kuhn, M. (1984), Mass budget imbalances as criterion for a climatic classification of glaciers, *Geogr. Ann.,* 66A, 229– 238.

Rasmussen, L.A. 2004. Altitude variation of glacier mass balance in Scandinavia. *Geophys. Res. Lett.,* 31(13), L13401. (10.1029/2004GL020273.)

*Supplement*
In the main text of the paper, provide much more explicit referrals to the supplement when needed.

*Stylistic/flaw/typos*
P5-L5. The "glacier" surface energy balance, not glacial

P9-L14. Therefore

P12-L6. Therefore

P20-L12.  Point to be deleted after "shown"

P21-L33. Add a space bar "52 kgm$^{-2}$ for winter"

---

## Short Comment (SC1) · 6 Mar 2017

**General comments**

The paper from Galos et al. presents a valuable contribution in the field of glacier mass balance monitoring, implementing reanalysis procedures optimized in the framework of the World Glacier Monitoring Service (Zemp et al., 2013). In particular, the field measurement efforts, the completeness of the reanalysis and the detailed quantification of uncertainties are appreciable.

Due to logistical issues and/or to the peculiar characteristics of monitored glaciers, it is often impossible to setup point monitoring networks that are evenly distributed and cover the entire surface of the glaciers. Therefore, there is usually the need for interpolating and extrapolating point measurements over unmeasured areas, which often have high lateral gradients of mass balance. In such circumstances, the analyst's knowledge of the monitored glacier becomes decisive, and greatly benefit from repeat observations of snow cover patterns during the ablation season, over several years. The Authors of this paper make extensive use of this information for mapping the mass balance distribution over Langenferner, which in my opinion is another added value of this work.

However, I have one point to highlight, which is the calculation of so-called pseudo-observations in the upper part of the glacier for years without direct measurements in that area. Galos et al. use a physically based energy and mass balance model to do that, starting from meteorological data recorded by weather stations located in the proximity of the glacier. There are some weak points on the way the model has been applied and the transfer function for meteorological variables have been calculated (see detailed comments); apart from this, my principal criticism concerns the use of mass balance models for calculating artificial point measurements, which are required in usampled areas for completing glacier-wide mass balance computations.

Because the high value of mass balance records lies, among others, in their useability as sensitive climatic indicators and for improved understanding of glacier-climate interactions, using pseudo-measurements modelled from meteorological data is a sort of circular procedure.

Previous works by e.g. Haefeli (1962), Jansson (1999), Carturan et al., (2009), and Kuhn et al., (2009) proposed extrapolation procedures that are independent from meteorological observations. In my paper on the mass balance series of the neighbouring La Mare Glacier (Carturan, 2016), I tried implementing these procedures for mass balance extrapolation, combining point measurements and snow cover pattern observations.

Therefore, I recommend at least mentioning these works in the Introduction section, clearly stating why a mass balance model has been preferred for mass balance calculations in unsampled areas, and discussing the limitations of this approach. In particular, I suggest discussing the generalizability of the method (e.g. for glaciers with few or absent meteorological data, indispensable for calculating transfer functions of meteorological variables, glacier cooling effect, etc), and the fields of application of mass balance series that are partly derived from meteorological data (ok e.g. for estimating contribution to sea level rise and management of regional water supplies, but maybe less reliable for early-detection strategies of climate change and process understanding).

References

Carturan L, Cazorzi F and Dalla Fontana G (2009) Enhanced estimation of glacier mass balance in nsampled areas by means of topographic data. Ann. Glaciol., 50, 37–46 (doi: 10.3189/172756409787769519)

Carturan, L. (2016) 'Replacing monitored glaciers undergoing extinction: a new measurement series on La Mare Glacier (Ortles-Cevedale, Italy)', Journal of Glaciology, 62(236), pp. 1093–1103. doi: 10.1017/jog.2016.107.

Haefeli R (1962) The ablation gradient and the retreat of a glacier tongue. In Symposium of Obergurgl, IASH Publication, vol. 58, 49–59

Jansson P (1999) Effect of uncertainties in measured variables on the calculated mass balance of Storglaciären. Geogr. Ann. Ser. A-phys. Geogr., 81(4), 633–642

Kuhn M, Abermann J, Bacher M and Olefs M (2009) The transfer of mass-balance profiles to unmeasured glaciers. Ann. Glaciol., 50 (50), 185–190 (doi: 10.3189/172756409787769618)

**Detailed Comments**

Page 2, Line 15-25: In the Introduction, I suggest mentioning published works dealing with point mass balance extrapolation methods that are independent from meteorological data (e.g. mass balance vs. altitude). See references in the general comments.

Page 4, line 15: extensive *net* accumulation measurement. Also in the following I suggest specifying net accumulation where required.

Page 4, line 26: I suggest writing here which time system is used (fixed date)

Page 5, line 26-29: did the Authors consider the possibility of using the bedrock topography, as obtained from geophysical measurements, instead of the surface topography, for identifying the divides?

Page 6, line 21: how did the Authors account for this problem? Here and in other parts of the paper the reader is left (temporarily) without explanations

Page 7, line 3: stratigraphic correction only for snow cover (is it fresh snow after measurement?) and not for ablation? How was it done?

Page 7, line 12: raw precipitation or (gauge-undercatch) bias-corrected precipitation? Possible impact on calculations?

Page 7, line 29-31: it is a fact that the spatial variability of the mass balance is large, and largely dependent on the spatial variability of micro-meteorological variables and local topography. Does the model fully account for these factors? Maybe the Authors should briefly recall here the main processes accounted for in the model. Moreover, physically based energy and mass balance models strongly rely on accurate spatially distributed fields (or in situ measurements) of several meteorological variables for their application. Because this is not the case of Langenferner, as stated at line 27 and in Section 3.2.1, there is the need for spatially flexible model tuning for the choice of the optimal parameters at the individual points, based on summer snowline observations. In my opinion, such use of a physically-based energy and mass balance model, instead of simple statistical relationships applied to measured point mass balances and observed snow lines (i.e. independent from meteorological observations, e.g. Carturan, 2016) is questionable and should be better motivated by the Authors.

Page 7, line 31 to Page 8, Line 2: I think that actual measurements, instead of pseudo-observations derived from climatic/meteorological data, should be preferably used in investigations concerning glacier-climate interactions (there is a risk of circular reasoning).

Page 8, line 7-8: fixed lapse rates and fixed offsets of temperature imply fixed glacier cooling/damping effects and simple air temperature distribution, whereas in the recent literature there is evidence of a significant variability of these processes within/among glaciers (e.g. Greuell and Böhm, 1998; Shea and Moore, 2010; Petersen and Pellicciotti, 2011; Petersen et al., 2013; Carturan et al., 2015). The same is valid

for the other meteorological variables, which have been calculated using transfer functions derived at one point over the glacier. I suggest at least discussing this issue.

Greuell, W. and Böhm, R.: 2m temperatures along melting midlatitude glaciers, and implications for the sensitivity of the mass balance to variations in temperature, J. Glaciol., 44, 9–20, 1998.

Petersen, L. and Pellicciotti, F.: Spatial and temporal variability of air temperature on a melting glacier: atmospheric controls, extrapolation methods and their effect on melt modeling, Juncal Norte Glacier, Chile, J. Geophys. Res., 116, D23109, doi:10.1029/2011JD015842, 2011.

Petersen, L., Pellicciotti, F., Juszak, I., Carenzo, M., and Brock, B.: Suitability of a constant air temperature lapse rate over an Alpine glacier: testing the Greuell and Böhm model as an alternative, Ann. Glaciol., 54, 120–130, 2013.

Shea, J. M. and Moore, R. D.: Prediction of spatially distributed regional-scale fields of air temperature and vapor pressure over mountain glaciers, J. Geophys. Res., 115, D23107, doi:10.1029/2010JD014351, 2010.

Page 8, line 24: I think the main reason for this improvement lies on a better calculation of the liquid vs. solid precipitation fractions. Do the Authors agree?

Page 8, line 28: $\Gamma_0$ therefore mainly accounts for gauge undercatch errors, precipitation vertical and horizontal gradients, and snow redistribution (and possibly ablation?). I suggest commenting on this.

Page 9, line 16-18: this tuning procedure is based on the assumption that the parameters controlling ablation processes, tuned at stake 22, are also valid elsewhere and that there are no significant biases in the calculation of spatial fields of meteorological variables, because otherwise there could be compensating effects from errors in modelling accumulation and ablation processes. My suggestion is to briefly discuss this aspect. In addition, I suggest better clarifying and repeating here for which stakes and in which years the model calculations have been done.

Page 15, line 31: is this assessment bases on the procedure proposed by Rolstad et al., (2009)? If so, I suggest adding this reference here.

Page 16, line 5. Compared to the period from 2005 to 2012, in 2013 the AAR was significantly larger in most glaciers of the Ortles-Cevedale. The effects on the mean glacier density were likely small or negligible, but maybe the Authors could shortly comment on that.

Page 17, line 26: maybe here the Authors could replace objective with automatic extrapolations. Here and elsewhere, specify net (accumulation) where required.

Page 19, line 23-24: in my opinion this is a key point and an essential prerequisite for reliable mass balance mapping and calculations. The Authors should mention here previous works highlighting the importance of observations regarding the snow cover pattern (e.g. Kaser et al., 2003; Østrem and Brugman, 1991)

Kaser, Georg, Andrew Fountain, and Peter Jansson. A manual for monitoring the mass balance of mountain glaciers. Paris: Unesco, 2003.

Østrem, G., and M. Brugman. "Mass balance measurement techniques." A manual for field and office work, Environment Canada, Saskatoon (1991).

Page 21, line 17-18: this good agreement is largely dependent on snow line observations, which have been used as constrains for model calibration at individual points, and on the representative location of stake 22 (where the model has been optimized) compared to the location of the modelled stakes. Given also the simplified transfer functions used for the meteorological variables, I wonder how much generalizable the proposed method is, what is its added value in comparison to statistical procedures based only on observations (point measurements and snow line mapping), and how to consider mass balance pseudoobservations derived from climatic data used alongside actual observations. In my opinion, the implementation of such combined (measured + modelled from meteorological data) mass balance datasets poses some limitations to their use for process-understanding or as key variables in monitoring strategies of the Earth climate (Zemp and others, 2005; WGMS, 2008). This is because the climatic indicator (glacier mass balance) becomes dependent on the same climatic variables it should be a proxy of.

Zemp M, Frauenfelder R, Haeberli W and Hoelzle M (2005) Worldwide glacier mass balance measurements: general trends and first results of the extraordinary year 2003 in Central Europe. In Data of Glaciological Studies [Materialy glyatsiologicheskih issledovanii], Moscow, Russia, vol. 99, 3–12.

WGMS (2008) In Zemp M and 5 others eds. Global glacier changes: facts and figures. UNEP, World Glacier Monitoring Service, Zürich, Switzerland

Figure 2: it is unclear if the snow probings refer to winter mass balance measurements or to annual net accumulation measurements. In the second case, the location of winter balance measurements should be added.

Figure 7: I suggest adding the 1:1 line

---

## Author Comment (AC1)

**Renalysis of a ten year record (2003-2014) of seasonal mass balance at Langenferner, Ortler Alps, Italy**

**Author's reply to referee- and short comments**

Stephan Peter Galos and Co-authors

April 18, 2017

**Introductory Remarks**

We thank the two referees, Liss Andreassen and Emmanuel Thibert for reviewing our manuscript and their thoughtful comments. We'd also like to thank Luca Carturan for his detailed short-comment.

The present reply-letter is the response to all three comments and is structured as follows: Section 1 provides detailed answers to concerns raised by the two referees and the author of a short-comment. Sections 2 to 4 then offer a point by point response to the remarks of each referee / short comment author.

The referee's comments are given in *grey italics* while our responses are in regular font. Unless not explicitly stated otherwise, cross-references in the text always refer to the sections of this letter and not to the paper. Reference-lists of the original referee- and short comments are omitted for readability. All references are summarized at the end of this document.

We want to mention that some numbers for annual and summer balances changed slightly ($\pm$ 3 $kg\ m^{-2}$) in the revised manuscript because the *reference method* was initially calculated with incorrect glacier outlines. All numbers concerned (mass balances, related statistics and uncertainties) in the text and the tables of the manuscript were changed and figures were adjusted where necessary.

**1 Author's response to important concerns shared by the referees**

**1.1 Mass balance measurements and data modeled from atmospheric input**

Both referees and the author of a short-comment address the use of modeled point mass balances in observational series which under certain circumstances may be problematic. We agree that glacier mass balance data based on meteorological variables have to be clearly flagged and implications such as a the dependence to atmospheric variables - as mentioned by the referees - have to be indicated and discussed in an appropriate matter. This was not the case in the discussion version of our paper manuscript and has been changed in the revised manuscript.

The potential of models based on meteorological data in modern glacier mass balance monitoring strategies has been outlined by a number of authors (e.g. Machguth et al., 2006; Huss et al., 2009, 2013). Consequently, during the past years many studies have made use of the benefit offered by a combination of direct measurements, observations and models. The series of published works includes the reconstruction of glacier wide mass balance series from sparse point measurements (e.g. Huss et al., 2015), the application of the model as an enhanced method for the spatial extrapolation of point measurements to unmeasured regions of the glacier (e.g. Huss et al., 2009, 2013; Sold et al., 2016), the extrapolation of measurements in time (e.g. Huss et al., 2009) and the integration of auxiliary data such as snow (line) observations (e.g. Huss et al., 2013; Barandun et al., 2015; Kronenberg et al., 2016; Sold et al., 2016).
Some of the mentioned works are explicitly presented as mass balance reanalysis studies (Barandun et al., 2015; Sold et al., 2016) and the model based techniques presented in these studies often enable the (currently) best possible estimate of glacier mass balance. This results from the fact that the performance of modern model approaches is generally superior to statistical approaches. This in turn is owed to the additional information (local topography and related implications on governing micro-climatological parameters such as temperature and radiation, snow line, snow depth and density, etc.) which is used to more accurately calculate the mass balance of a certain point or glacier.
However, the use of meteorological data to drive the models is regarded as problematic since observational series of glacier mass balance are used in investigations of glacier behaviour in response to climate forcing, which in turn is represented by atmospheric parameters. Therefore these data must be flagged and information on how the data was compiled must be provided in a fully transparent matter to enable possible data-users to decide whether the data suits their purpose or not.

In order to eradicate shortcomings in the original manuscript related to this topic we have included the following improvements in the revised version of the paper:

- The revised paper contains a clear indication of the problems associated with integrating meteorological variables in observational series of glacier mass balance

- We present a stronger motivation for the choice of the presented model approach and explain why we did not rely on another (statistical) approach such as the Lliboutry model

(Lliboutry, 1974) or other techniques mentioned by the two referees and the author of a short comment.

- We included additional references dealing with mass balance reconstruction based on statistical methods

- We critically discuss the implications of using a model approach like ours in the results section and comment on limits in the transferability of the presented method

- We changed Figure2 and Table1 in a way which makes it easy to see which years and seasons are influenced by modeled values and to which extent.

The reanalized mass balance data (point values and glacier wide averages) are meanwhile submitted to the world glacier monitoring service (WGMS). Thereby all data which are fully or partly based on modeling are clearly flagged.

**1.2 Mass balance modeling**

Referee Nr.1 and the author of the short-comment express their concern regarding the applied model approach with detailed comments addressing model properties/tuning, meteorological input and the transferability of the approach. Here we would like to emphasize that the goal of our study significantly differs from that of a modelling study. While energy balance models are usually applied for studying the interplay between atmosphere and glacier surface, our study does not aim to resolve physical processes behind glacier energy and mass balance. This strongly relativizes many of the concerns about model properties, extrapolation of meteorological model input or the fitting procedure of the precipitation scaling factors.
The goal of this study was to use the model as a tool to calculate the mass balance at a given point as realistically as possible. The validation of the model output against observations yields uncertainties which are markedly lower than those of other methods and hence gives us confidence that our goal was reached.

Without starting a flame war against statistical approaches, it can be argued that statistical methods relying on simplified spatial and temporal relationships (such as a temporally constant spatial mass balance gradient) suffer a number of drawbacks. Statistical methods damp the spatio-temporal variability of mass balance as they imply (linear) correlation between mass balance points or glacier parts or other simplified assumptions. As such they must be handled at least as carefully as series partly based on points modeled from atmospheric input when it comes to process oriented applications.

In the revised manuscript we discuss the above mentioned issues more explicitly to motivate our choice for the physical model. Possible restrictions in the applicability of the approach to other sites and for other purposes are discussed in the text of section 5 (Results and discussion) of the revised manuscript.

**1.3 Glacier mass balance units**

Both referees argue against the use of the unit $kg\ m^{-2}$ in our paper and suggest $m\ w.e.$ instead. We would like to keep $kg\ m^{-2}$ for the following reasons:

- $kg$ (mass) and $m$ (length) are both part of the SI-unit system while "water equivalent" (*w.e.*) and $mm$ are not

- $m\ w.e.$ often requires at least two, in our case even three decimal digits which makes it uneasy to read

- $kg\ m^{-2}$ is easy to read and convenient since it is equivalent to $mm\ w.e.$ which is familiar to any glaciologist

- $kg$, as a unit of mass, seems appropriate for a mass-balance study

- Finally, we follow (Cogley et al., 2011, p13, lines 37 to 40), who wrote : *"Diverging from Anonymous (1969), we have accorded primacy to mass units rather than volumetric units. This means that, although we do not discourage use of the metre and millimetre water equivalent, we consider that usage and understanding would be the better for a stronger emphasis on the difference between mass and volume".*

It is also suggested to present annual and seasonal mass balance as a function of time. Here again we refer to Cogley et al. (2011, p.66, lines 1 to 8). The unit for specific (per unit area) mass balance is $kg\ m^{-2}$ while $kg\ m^{-2}\ yr^{-1}$ refers to a mass balance rate. The use of mass balance rates is appropriate in cases where the presented rate is valid for a longer time-span (longer than the unit of time used to give the rate itself). Hence, mass balance rates should be used when presented numbers are averaged over several years, or where they are regarded to be valid over a longer time period. This is not the case here since we present mass balances for individual years.

**2 Point per point reply to referee-comment Nr.1 by E. Thibert**

**2.1 General Comments**

*S. P. Galos and co-authors provide a data reanalysis scheme applied to a ten year record (2004 to 2013) of seasonal mass balance at Langenferner glacier in the European Alps.*
*The approach involves homogenization of available point values and reconstruction of missing data for years and locations without measurement by the application of a process-based model constrained by snow line observations. Point mass balances are then extrapolated to the overall glacier surface to quantify the glacier-wide balance help to different extrapolation techniques and recourse of topographic data. The 2 reconstructed seasonal series differ notably from the original records. The new annual mass balance series is compared to long-term volume changes determined from airborne laser-scanning data in a rigorous error analysis and following the framework proposed by (Zemp et al., 2013). The authors find good agreement between those determinations, the residual discrepancy being explainable by the natural scattering of the data. These favourable results are to confer a significant confidence in the correctness of the re-analysed glacier-wide mass balance times-series of Langenferner glacier.*
*The paper employs advanced methods of mass balance computations, and geo-statistical inferences. The paper is clear, well organized, and properly focuses the scope of the journal. I think this paper is to be welcomed also as a new appropriation from the glaciological community of the guidance proposed 4 years ago by (Zemp et al., 2013).*

*I have two main questions regarding the model tuning for point balances, and the spatial/temporal pattern of balances:*

*1°) I find the calibration of the accumulation rather weakly constrained on how the precipitation scaling factors $\Gamma_{i,a}$, are allowed to vary spatially (from stake-to-stake) and with time (from year-to year).*
*In the present formulation, $\Gamma$ is allowed to account for any deviation from the observed precipitations at all stakes and independently from what happens at other stakes. The authors mention on line 25 that the spatial pattern of $\Gamma_{i,a}$ is reasonable regarding terrain curvature and wind effects. Analysing Figure 5, it seems also that some stakes systematically have lower $\Gamma$ (stakes no. 21 or 25) and on some years all stakes deviates accordingly up (2009, 2011) or down (2010, 2004). Can the authors provide a colour coded map of the mean spatial structure of $\Gamma$ between 2004 and 2013 which could sustain this? Therefore one should expect a much more constrained formulation in Equation (2) when it comes to tune the model at individual stakes. Maybe the authors could reformulate $\Gamma$ as $\Gamma_{i,a} = \gamma_i \, \gamma_a$, fitting 2 uncorrelated functions holding spatial and times dependencies of accumulation, and accounting for systematic deviations at some locations ($\gamma_i$) and on some years ($\gamma_a$).*

We prefer not to further constrain this parameter, as we are not sure what the added value of this procedure would be. A stronger constraint would limit the allowed range of $\Gamma$ which in turn would have negative impact on the model skill. We see this tuning parameter as an "inverse model", allowing us to reconstruct the mass-balance from observations of the ice (or firn) emergence. In the revised manuscript we commented more clearly on the fitting procedure of $\Gamma$ and related implications for the model approach.

The role of the scaling parameter Γ is to account for local accumulation characteristics such as snow redistribution, which are strongly variable in space and time and are a priori unknown and independent from the characteristics at other (measured) stake locations. While the primary goal of the paper is not to learn about these processes, the spatio-temporal variations of Γ indeed have a lot to say about them.

As suggested, we added a color coded map to the supplementary material showing mean Γ at the locations to which the model was applied. In the same figure we also added a colour coded map of the 2005 winter balance showing that high (low) values of Γ coincide with high (low) values of local winter mass balance.

In principle we appreciate the author's suggestion, as it would provide a new way to assess the applicability of the "Lliboutry model" (the suggested constraint is very close to the underlying assumptions of Lliboutry's model). We plan to compare these strategies (Lliboutry, Contrained Γ, Free Γ) in the frame of a master thesis at ACINN.

*2°) Figure 6 shows that the altitudinal profiles of balance and the way they change over time somewhat follow the time-space decomposition proposed by (Rasmussen, 2004) or (Kuhn, 1984):*

$$b(z,t) = f(z) + \delta b(t).$$

*In search of this, the authors could test if there is strong correlation between readings at the individual stakes. This feature also suggests that Lliboutry's model should work on the data set. A 10-year time series is just long enough and eminently suitable to test Lliboutry's linear model. I would therefore encourage to possibly test this analysis all the more that it could provide estimates for missing values of point balances in the accumulation area at the beginning of the record. Moreover these estimates would be free from any meteorological or glacier-to-climate assumption and safeguards the reconstructed series for an independent and unbiased analysis with climate drivers.*

Again the referee's suggestion is valuable and interesting but goes beyond the scope of our paper. Lliboutry's model is a powerful tool to reconstruct mass balance values for locations and years without measurements. Nevertheless we have decided to use a different approach for several reasons: First the temporal series of several of the stake locations for which we apply our model are quite short. Some do not comprise more than three or four measured years and hence the local characteristics ($\alpha_i$ (e.g. Thibert and Vincent, 2009, equation 3)) are quite unreliable since loosely constrained. Apart from this the reported uncertainties for individual point mass balances calculated using the Lliboutry model are notably larger than the uncertainties of our approach which for individual points and years are in the order of 129 $kg\ m^{-2}$ (Figure7 in our paper). However, a direct comparison of the methods is outstanding and could be an interesting perspective for future research.

Please see our detailed statement in section 1 for changes which were included in the revised manuscript.

*Here follow some detailed questions, comments, suggestions, and indications of minor typos in the paper. Please consider them in the same positive way in which they are proposed.*

**2.2  Specific Comments**

*Substantive comments*
*P1-L1. You could mention that both uncertainties (scattering) and incorrectness (biases) affect calculations.*
We clearly describe the composition of uncertainties and errors, as well as their origin in section 4 of the paper. As the abstract should be short, concise and well readable it offers limited possibilities for explanations and we would prefer not to add this detail in this part of the paper.

*P1-L11.  Annual values for mass balance refer to changes per unit of time (rate) and should here and throughout the manuscript expressed per year ($yr^{-1}$). Although there is not uniformity of units regarding specific mass balance (per unit of area), the authors could use the convenient, numerically equivalent and more meaningful mm w.e. or, alternatively, the SI unit m w.e. instead of $kg\ m^{-2}$.*
Please see our detailed statement in section 1.3

*P1-L19.  Don't you think that a more direct (physically-based) driver is rather the surface energy balance?*
Although the surface energy balance of glaciers enables understanding of many processes behind glacier changes it does in the end not provide information on whether a glacier is losing or gaining mass because it does not allow conclusions on accumulation processes. In contrast, the (surface) mass balance of a glacier accounts for both accumulation and ablation processes - which are both equally important for glacier behaviour - while the energy balance mainly allows to deduce ablation processes.

*P3-L21. Again please express rate units per year ($yr^{-1}$).*
Please see our detailed statement in section 1.3

*P4-L10. You mean that these stakes couldn't be measured any longer?*
Yes, exactly. The locations became free of ice during the study period. We changed the sentence to clarify.

*P4-L26. Please use when relevant "glacier-wide balance" terminology to fit Cogley et al.(2010) glossary.*
In this special case the terminology is OK since we did not only submit glacier wide balances but also point balance and mass balance versus altitude (annual and seasonal specific mass balance for individual 50 m altitude bands) information. However, we generally tried hard to follow Cogley et al. (2011) wherever possible and therefore clarified the terminology wherever we felt that there could be a flaw.

*P5-L24. A bit more needs to be say on how single out definitely areas outside the glacier subject to elevation loss (erosion in moraine terrain) and area on the glacier subject to reduce ablation (thick debris)?*

Debris cover and related problems/processes play a minor role at Langenferner since only a few percent of the glacier is debris covered. Especially for the glacier wide mass balance the influence of debris cover is negligible. However, we briefly discuss this point in the revised manuscript and added an orthophoto-map to the supplementary material of the paper where (the small) debris covered areas are indicated.

*P7-L19. A 10-year time series is just long enough to test Lliboutry's linear model as the spatial terms generally converge to steady values over this time scale.*
For many of the stake locations where the model was applied to we only have five years of measurements, for some even only three or four. Please see our statements in section 1 and our explanations regarding point 2°) of section 2.1.

*P8-L15. At which spatial scale does ERA-Interim provides time series for atmospheric pressure?*
"The spatial resolution of the data set is approximately 80 km (T255 spectral) on 60 vertical levels from the surface up to 0.1 hPa." from: http://www.ecmwf.int/en/research/climate-reanalysis/era-interim.
We added the distance of the nearest grid-point from the glacier to the paper text.

*P8-L26 to P9L8. How ranges the pre-calibrated scaling factor $\Gamma_0$ compared to the altitudinal gradient of precipitations?*
We did not investigate this explicitly since it is not relevant for the paper (see our statement addressing 1°) in section 2.1). But when having a closer look to the range of $\Gamma_{i,a}$ it is obvious that $\Gamma$ is determined by accumulation re-deposition (mainly due to wind) rather than an altitudinal precipitation gradient. We all know this also from field observations where certain locations "always" show very shallow snow cover or even bare ice while other locations are burried by meters of snow. To a certain extend such patterns re-occur every year, but still there can be differences depending on absolute snow amount, prevailing wind directions, storm events, timing and strength of melt or rain events, etc.

*P9-L10 to 28. As mentioned above, the calibration is very weakly constrained on how $\Gamma_{i,a}$ is allowed to vary spatially and with time and one should expect a much more constrained formulation in Equation (2). Please refer to my general comment ahead.*
We did refer to the referee's general comment in our statement addressing 1°) in section 2.1.

*P10-L29. I suspect a 1x1m resolution grid to be an oversampled interpolation. Considering 250 kg m-2 contour-lines and analysing Figures 4 and 6 suggests that just 4 to 5 contour-lines should cover the altitude range [3125-3375 m] over nearly 1 km in the accumulation area. Which means that a 100x100 m would be the right sampling scale adopting a maximum of 2 grid cells between contourlines.*
We used the high resolution of 1 m because this enabled us to accurately derive glacier outlines/margins. All calculations in the paper, except for the boot-strap uncertainty calculations (where a 5 m grid was used for reasons of computational effort), were performed using a 1x1 m grid resolution. This was also done as the surface mass balance grids served as input for the calculation of glacier outlines in years without ALS or orthophoto data (paper section 3.5). For this application a high resolution is a precondition. Of course a 1m resolution grid is subject

to noise but (i) due to the high point density of ALS measurements this noise is limited and (ii) for spatial integrations stochastic errors related to the high resolution do not have an impact. When smoother grids are needed for whatever application they can easily be resampled from the 1 m raster.

The density of contour lines can by the way not be deduced from Figure 6 since the Figure shows mass balance averages over 50 m altitude bands which does not say anything about local mass balance gradients resolved in the manual 2-D analyses. We added a Figure to the supplementary material showing the contour lines used to calculate the mass balance. We are aware of the fact that the high degree of resolved details is not necessary to calculate spacial averages but it may be the basis for studies on physical processes related to surface mass balance and its spatial distribution.

*P11-L25. Don't you think it should be better to adjust field data to preserve the independency of the geodetic balance as a reference control?*
Yes, of course those values should be preserved, but this is also valid for the field data. Therefore we applied the correction only to make a comparison between the direct and the geodetic methods possible and reasonable. The original values for the time period between the ALS campaigns without any corrections are presented in Table 3 and in the text of section 5.4 of the paper just besides the corrected ones. However, we included the value for the snow (density) corrected results in Table 3 to have all three relevant numbers (uncorrected, corrected for snow, corrected for snow and time difference) present in the revised manuscript.

*P13-L15. Looks like my comment on P5-L24 on how single out areas outside the glacier subject to erosion and area on the glacier subject to low ablation.*
Please see our comments concerning the similar issue on P5-L24.

*P15-L5. Here and in section 4.2, I am not sure if 3 levels of subsections are worth.*
The comment is understandable since the individual sub-sub-sections are quite short. But we chose this structure to make it more easy to follow and for someone who may use this part of the paper as a guideline, it is easier to follow since each of the sub-sub-sections describes one of the terms in equations 14 and 15.

*P14-Equation 14. Here you assume errors to be uncorrelated to combine them quadraticaly. To some extent, this general formulation from Zemp et al. (2013) does not account that errors at a point and errors associated to the spatial integration are combined through weighting coefficients in the spatial averaging. Make here an explicit referral to the supplement S2.1 where details can be found about error calculations.*
Done. We also added a comment on this in the revised version of the paper since this issue is something which many people seem to disregard.

*P16-L12. Unit $kg\ m^{-2}\ yr^{-1}$?*
No, not per year. This value is independent from the duration of the period of interest. It refers to the uncertainties related to differences in snow cover between the two ALS-acquisition dates and to differences in survey dates between the glaciological and the geodetic method. These uncertainties result from the inaccuracy of assumptions related to the described issues and affect the geodetic balance over a whole period.

*P17-L5. How do you think your re-analysis model will perform in case of systematic positive balances over few years? Would you have to recalibrate it? Or to reformulated the approach?*
As stated in the manuscript, in case of clearly positive mass balance the approach cannot be applied in the presented way as it relies on information on the date of ice (firn) emergence. However, it could for instance be applied to extrapolate snow measurements in time and to calculate the fixed date balance based on a measurement performed at a random date in summer. We do not think that the performance of the model is dependent on the absolute value of the mass balance at a certain point but rather depends on the amount and strength of summer snow fall events (with temperatures around the threshold distinguishing liquid and solid precipitation) between the measurement taken and the end of the hydrological year or the reference date to which the mass balance information should be extrapolated by the model.

*P17-L8. Title section 5.2: Mean (specific) glacier-wide annual balance Specific means per unit area, not averaged at the glacier scale.*
According to Cogley et al. (2011, p63, specific mass balance) this is not totally correct. Although we use the term *"specific mass balance"* as proposed by the referee, we have to point out that *"mean specific"* can very well be used for mass balances averaged over the glacier area. However, we changed the title to "Glacier wide specific annual mass balance".

*P17-L9. Can you estimate the spatial variability of the mass balance at the glacier scale from the five different extrapolation methods? And for winter and summer balance as well?*
We are not sure whether we correctly understood the question. The spatial variability of surface mass balance is of course best reflected by the contour line based methods. This issue is discussed in paper section 5.2 and indirectly in paper section 3.3.1 where the benefits of the contour line method are listed. Apart from that , the total range of observed point balances (max-min) is preserved by all methods, except for the profile method which tends to underestimate total variability.
We added a statement to section 5.2 of the revised paper version which discusses the matter of spatial mass balance patterns more clearly. We did not include a quantification of differences between the mass balances calculated from the different methods at a certain unmeasured point or area since this goes beyond the scope of the present study.

*P17-L20. Biases units kg $m^{-2}$ $yr^{-1}$?*
Please see our statement in section 1.3.

*P20-L10. It would be helpful to remind the reader that you did not correct for such internal processes.*
Done, we inserted a reminder a few lines earlier in the second sentence of the subsection.

*Tables*
*Table 4. Legend. Remove "s" after δ.*
Done.

*Figures*
*Figure 6. I don't find where the main text refers to Figure 6*

It is in section 3.5 where we refer to "Fig. 4 and 6".

*Figure 7. I suggest to add the 1:1 line and the fitted line with the equation terms providing intercept (bias) and slope.*
Done.

*Figure 10. Is it possible to plot error bars (one sigma) on cumulative balances.*
In the three upper panels we did it for geodetic and and reference method. In the lower panel we tried out plotting the accumulated random errors as calculated in paper section 4 but they were not visible since uncertainty bars are smaller or at the end of the period only slightly larger than the markers (less visible than in the rightmost of the upper sub-plots for 2011-2013). For style reasons we omitted them.

*Supplement*
*In the main text of the paper, provide much more explicit referrals to the supplement when needed.*
Done. Please note that the supplementary was substantially improved in the revised version.

*Stylistic/flaw/typos*
*P5-L5. The "glacier" surface energy balance, not glacial*
Thanks, ⇒ done.

*P9-L14. Therefore*
Changed.

*P12-L6. Therefore*
Changed.

*P20-L12. Point to be deleted after "shown"*
Done.

*P21-L33. Add a space bar "52 kg m$^{-2}$ for winter"*

Done.

**3 Point per point reply to referee-comment Nr.2 by L. Andreassen**

**3.1 General Comments**

*The authors reanalyze a 10-year record of mass balance of a small glacier in the Italian Alps. They use previously applied approaches combined with new approaches. They present a thorough error calculation. The methods and results are in general carefully described. They make use of pseudo-observations to reanalyze the data and calculate the glacier wide mass balance. This study is thus not only a modelling experiment, but presented as a reanalysis of mass balance providing updated series of glacier mass balance. Such mixing of modelling and field-data is not unproblematic, but the authors do not discuss this. I miss a section in the paper discussing this choice. Furthermore, I miss a paragraph where the authors discuss the implementation of their results and flagging of the series.*

The referee's remark is correct. Please see our detailed statement in section 1.

*Information on number of pseudo stakes used for every year and which years that used modelled pseudo points in the calculation is lacking from the table. If some years need to be modelled and not others, this should be more clearly stated and flagged in the paper and resulting tables. A comment field in the table could be useful to so the series are marked with reanalysis and comment on modelling degree.*

We present an improved Table 1 which focuses more on the differences in glaciological variables between the original record and the reanalized series and providing information on the degree of modeling involved in the mass balance of individual years and seasons.

*Fig 2 could be extended with all 10 years so the data sources are shown.*

We followed this suggestion by changing Figure 2 to a nine-year panel showing the data sources for the years 2004 to 2012. The set up for the annual mass balance 2013 can now be seen in Figure S2 of the supplementary material.

*Data availability*
*According to TC journal instructions, "Authors are required to provide a statement on how their underlying research data can be accessed". A section on Data availability therefore needs to be added to the paper with information on where to find the underlying data. The World Glacier Monitoring Service is the appropriate data service for much of the glaciological data.*

The required statement was added and the data has been submitted to the world glacier monitoring service.

**3.2 Specific Comments**

*Units. I suggest using m w.e. or m w.e. a-1 instead of kg/m2. Alternatively, mm w.e. Kg/m2 is not standard in the literature.*
Please see our detailed statement in section 1.3.

*P2. L30. Very long sentence.*
The long sentence was split in two.

*P3. Study site and data could be divided in 2 chapters.*
Well, this is true. But on the other hand they are not so easy to separate since some information (e.g. section 2.2 Glaciological measurements) which is relevant for the data is strongly related to the study site and vice versa. As a compromise and for a better overview we changed the structure by inserting a subsection named "Langenferner" which comprises all information which is just about the glacier itself.

*P3. L27. Could mention the mean value for the period.*
Please note that the paper is already quite long and for this reason the interested reader is referred to the references cited in the paper where she/he can find further information. Nevertheless, we changed the sentence and included the information as suggested by the referee.

*P4.'The current stake network October 2016' is not that relevant here as the study period is up to and including 2013. Rather give the information for 2013. In 2.1. or in the introduction objectives (P3) the study period could be mentioned explicitly.*
We followed both remarks. We changed the information on the stake network and included information on the study period in the introduction.

*P4. 15. Add 'of remaining snow from the previous winter' or something similar to be clear to not mix up with winter balance/accumulation. Same for L18. Could mention how large area (in %) that had snow remaining in 2010 and 2013.*
We included information about the areal extent of accumulation areas in both years but the accumulation area at the end of the hydrological year does not necessarily refer to areas with remaining snow from the previous winter. Instead, it refers to areas with positive mass balance. Especially in the year 2010 large areas in the upper parts of Langenferner had undergone only moderate ice melt in July and were covered by a relative thick layer of snow in August and September. This led to a positive fixed date mass balance at those locations although no snow of the previous winter could remain throughout the summer there. Hence, the proposed phrase "remaining snow from the previous winter" is in this special case not appropriate and would introduce misinformation instead of clarity. We used the term "annual net accumulation" to avoid confusion with winter accumulation or accumulation at any date other than the end of the hydrological year.

*P4. L26. Mention the first mass balance year reported, e.g. 2003/2004, it may not be very clear since measurements began in 20002 (L5).*
We added "since the beginning of measurements in the year 2003/04." to the last sentence of the paragraph.

*P5. L12. On 'Around', I assume also on the glacier since this is not specified. Replace around with covering or take out and just state the point density. Would it be better to resample to a 5 m model? The section 2.3 was quite brief.*
Yes, the mentioned point density also refers to the glacier itself. We changed the phrasing to

"on and around Langenferner".

Indeed the section is quite brief. But again the reader is (now more clearly) referred to another study where the DTMs are described in more detail if this kind of information is regarded as relevant. More info could be added in case of more specific comments on that.

DTM-resolution: We used the high resolution of 1 m because this enabled us to more accurately derive glacier outlines/margins. All calculations in the paper (except for the boot-strap uncertainty calculations where a 5 m grid was used for reasons of computational effort) were performed using a 1x1 m grid resolution. This was also done as the surface mass balance grids served as input for the calculation of glacier outlines in years without ALS or orthophoto data. For this application a high resolution is beneficial. Of course a 1m resolution grid is subject to noise but (i) due to the high point density of ALS measurements this noise is limited and (ii) for spatial integrations stochastic errors related to the high resolution do not have an impact. If a smoother 5 m grid is needed for whatever application it can easily be resampled from the 1 m raster.

*P5. GPS should probably replace with GNSS (Global Navigation Satellite System) throughout the manuscript.*
Done.

*P5. L23. How much of the glacier is debris covered? This is not stated here or in chapter 2. A picture or aerial photo could have been a nice addition to the paper. Orthophotos are available according to the text. Could be added in an improved figure 1. The section and the referencing is a bit unclear to me, was the approach done following Abermann et al (2010) or was is it done by Abermann et al (2010).*
Debris Cover: Only a minor fraction of the glacier's lower part is debris covered.

We added a comment in the text and referred to the supplementary material where an orthophoto-map was added which gives information on debris cover.

Orthofoto and Figure 1: Figure 1 was improved according to your suggestions but we did not include an othophoto as when we tried including an OF as a base map the result was an overloaded figure. Instead, we added an orthophoto-map of the glacier to the supplementary material of the paper showing the measurement set-up for the annual mass balance 2013 which was cut from figure 2 for reasons of readability (Panel of 9 years better fits to a whole page figure). This map also provides information on debris covered areas (indicated by red arrows).

Method and referencing: Both citations refer to the same paper (Abermann et al., 2010). However, the two (somehow confusing) respective sentences were changed for clarity.

*P5. L28. Changes in ice divide is negligible. Could this be quantified?*
It depends on if you regard the surface topography as a satisfying proxy for the ice flow direction. Differences between the DTMs of 2005, 2011 and 2013 are quite small (order of meters) and could also be due to differences in snow cover (2013). Hence, we did not discuss this issue in detail as it is not in the focus of the current paper.

*P6. Ch 2.5. Could the extraction of snow line information be illustrated. Maybe add a table to the supplementary material? How were the imagery geo-referenced and stakes identified on the imagery? Some more information could be added.*

As stated in the paper this was done manually without any geo-referencing of the images. The glacier is quite small and not too homogenous, consequently the location of stakes can quite easily be identified by someone who knows the glacier well. Nevertheless this issue is an important one, also limiting the 1:1 transferabilty of the method to other sites. Therefore we added a more critical discussion of this point in the results section and added an explaining section to the supplementary material.

*P6. L21. Write what you did instead of 'we accounted for this problem' to be specific.*
Done.

*P6. L28. This issue → this melt ...L29. Respectively → for the additional melt* We think that the first case is a matter of writing style and it is clear for the reader that the word "issue" is referring to the problem of ice melt. In the following sentence "respectively" was replaced by "late autumn ice melt".

*P7.L2-3 Unclear, do you mean that 'Whereas measurements were carried out close to the ... and thus reported as fixed date in the original data, the....*
The sentence was changed for clarity.

*P7. L10. When correcting accumulation: How are melting episodes accounted for?*
Melting epsodes were not accounted for. They only have to be accounted for if the melt water drains from the glacier during the correction period (i.e. between April $30^{th}$ and the date of winter balance measurement in early May) which at Langenferner is hardly ever the case in that time of the year. Snow pits dug for winter balance measurements indicate that the snow pack was never saturated with melt water, except for the extraordinary year 2007 when the snow pack in the lower glacier sections was saturated due to very warm weather in April. But even in that year conditions in early May were colder again leading to the assumption that even in 2007 melt water run-off can be neglected in this regard. Note that the applied correction method only corrects for the date difference between the end of the hydrological winter (April $30^{th}$ and the date of measurement). The reason for larger delays in measurements has always been unfavourable weather which in early May at altitudes of more than 2700 m (terminus of Langenferner) is commonly related to snow fall rather than significant melt.
However, we added an explaining sentence to the respective subsection in the paper and referred the reader to the supplementary material where we explain the assumption and resulting restrictions in the transferability of the method.

*P7. L17. L 21. L23. Is → was*
Changed according to reviewer's suggestion.

*L32. Add that is also justified by available data to run the model (this is not always the case)*
A statement was included in the improved motivation section.

*L29 Generated (use past tense on work carried out in this study)*
Probably referring to L19 (not 29)...changed according to referee's suggestion.

*L1. Provide to whom? 'Instead of ...only' → 'in addition to glacier wide balances' One must be care-*

*ful not mixing measured and modelled point data as this can add confusion, data should be clearly marked.*

Done. Text changed according to reviewer's suggestion. The issue of data flagging is now clearly discussed in the paper (see also section 1of this document).

*L33. What does 'After the individual extrapolation of point measurements mean here? Interpolation could be a better word.*

This point is not that trivial. While 'interpolation' refers to space or time between known (measured) points, extrapolation refers to space or time outside or beyond data points. In the case of mass balance calculation we actually deal with both (Zemp et al., 2013). Hence, the truth is actually that neither interpolation nor extrapolation is the fully correct term. It is true that the *topo to raster* tool is often called an interpolation tool, nevertheless it is also used to extrapolate data values for regions outside the measured area. Another example: For annual balance we often rely on firn density from one snow pit. In this case the density is definitely not interpolated. Similar is true for other situations.
As neither the one nor the other term is fully correct, we decided to go on with "extrapolation" in the revised paper for reasons of simplicity.

*L26. Mean manually drawn? Add this information.to be clear.*
Changed according to referee's suggestion.

*L28. What does the latter method refer to here? Specify the method to be clear.*
The method specified as 'the latter' refers to the last method described in the previous sentence and it is obvious from the content which method is resulting in rasters. For this reason we kept the sentence unchanged.

*P11. Suggest to call this paragraph be called interpolation. Usually one refer to interpolation methods in general, not extrapolation methods. E.g., the Topo to Raster tool is an interpolation method*
See above comment on the use of interpolation versus extrapolation.

*P11. L15. Why the e.g. reference here, unclear.as follows (Zemp et al., 2013) → following Zemp et al. (2013).*
Both issues changed.

*P11. L19. Is → was*
Changed.

*P13. L13. Serves → served*
Changed.

*P14. L29. Substitute 'the above problem' with 'uncertainties in points'*
Replaced by "the propagation of point scale uncertainties".

*P14. L10-12. Rough surface topography may give errors when measured differently by field observers. Uncertainties in identifying summer surface gives also uncertainties in point data.*
Reading errors and errors related to surface roughness are treated separately in our study. From time to time we perform reading experiments with student groups. Even in areas with

high surface roughness typical reading errors are in the order of 3 cm. Note that errors are expressed as standard deviations and that larger values can occur but are not too common. Uncertainties in identifying the summer surface are discussed in the paper. In our case they are only concering snow probings since in snow pits the summer horizon is easy to identify due to the very negative mass balances and the strong surface melt and resulting dirt accumulation at the surface throughout the study period.

*P14. Line 29. Instead of starting a subchapter 'Similar to the above problem', state what the problem is.*
Done, we changed the sentence.

*P15. L10 due to the fact that it.. → as it..*
Changed.

*P15. 4.2.1. Could give some more details on the coregistration.*
We changed the subsection by adding all information relevant for the paper.

*P17. L5. Specify why it could not be used.*
Done.

*P20. L20. Than → as*
Changed.

*P18. Line 8. Are the original ELA and AAR values changed from the original record? Please make a comment addressing changes in ELA and AAR and add the original values to Table 1 if changed.*
Yes, also the ELA and AAR of the reanalized series changed. This is now visible from an improved Table1 which - following the referee's suggestion - puts more focus on changes between original and reanalized reference series, as well as information on degree of modeling involved in the mass balance of the respective season or year.

*P19. L14. Add the value found by Galos et al. (2015).*
Done.

*P19. L15. Could add some results in the text for the short periods, not just referring to the table. Comment on the density conversion factor and uncertainty used for the short period, as short periods (1-3 yr) may have a different conversion factor. (Huss, 2013)*
We added results for the shorter periods.
Density conversion: we explained in detail how the geodetic mass balances were calculated. Thereby we also explain that we distinguish between fresh snow, snow from previous winter and ice. We applied the density conversion factor of 850 kg m$^3$ to the ice part of the glacier and used different densities for fresh snow and snow from the previous winter. The changing conversion factors for short observation periods as discussed in Huss (2013) refer to calculations where different densities (of snow and ice) are not explicitly accounted for.

*P20. L4. What do you mean with further calibration, do you consider some of the presented work as calibration?*
Not in the sense of mass balance calibration. We removed "further".

*P20. L20. Than → as*
Changed.

*L21. Specify by adding 'reanalysed' summer balance values?*
Done.

*L23. Add 'total', thus 'a total value*
Done. Also changed two sentences later.

*L25. Specify what you mean with individual contributors.*
Done.

*P20. L28. Could add 'reduced discrepancy' after recalculating.*
Done.

*P20. L31. Name the two methods. What does it mean not acceptable.*
Changed to: 'do not fulfill the [...] criteria'

*P21. L1. Specify where you recommend it to be used.*
Since the referee did not specify what she means with "it", we just assumed that "it" refers to the method in general. We added a more critical discussion on the applied approach to the revised version of the paper commenting on implication and limits regarding transferabilitya and possible alternatives.

*P21. L4. Is it not the glacier wide averages that are calculated based on pseudo observations? Check L2-5.*
We changed the sentence to make it clearer.

*P22. L8. As well as related → of. Inter-annual mass balance variability has indeed been considered before. Suggest rewriting.*
We do not state that inter-annual mass balance variability has not been addressed before but we changed the sentence.

*Figure 1. Add source and year of glacier outlines. The map should be fitted to either one-column or two-column width. Could be extended in east to allow some more space around Weissbrunnferner. Borderline on inset could be refined and the inset would probably look better without the shaded background.*
Source and year of glacier outlines: Done.
Figure width: The width of figure 1 is two column. For the submission of the manuscript we used the latest LATEX-template provided by *The Cryopsphere* where the figure and table widths are default for both one-column (8.3 cm) and two-column figures (12.0 cm) and tables. In the final version the sizes will meet the Journals requirements.

*Figure 2. Same comment as 1, could be extend to 2-column width. The dense outer outline could be thinned.0,5 → 0.5 Suggest to add all years here.*
Figure width: see above comment. More years: We have added all years from 2004 to 2012 and show the 2013 set-up in the supplementary material.

*Figure 7. Add 1:1 line Figure 8. Add (orig) and (ref) to the text.*
We added the 1:1 line as suggested. "orig" and "ref" is not appropriate here since modelled point values which were used for this validation are not used in the reanalysis. The shown validation is only possible for years and locations with modelled and measured values and in such a case the measured values were given priority in the reanalysis.

*Table 1. Add ELA and AAR if different from the original record. Add Area used in the original record. Add a comment field stating if modelled pseudo stakes are used for each year. Suggest having an additional table stating the data source available for each year including pseudo stakes used.*
Table 1 was changed following the referee's suggestions.

*In general, there are numerous 'in order to' in the manuscript. 'in order to' can be replaced by 'to' in many (if not all) occurrences.*
We replaced some of them.

**4 Point per point reply to short-comment Nr.1 by L. Carturan**

**4.1 General Comments**

*The paper from Galos et al. presents a valuable contribution in the field of glacier mass balance monitoring, implementing reanalysis procedures optimized in the framework of the World Glacier Monitoring Service (Zemp et al., 2013). In particular, the field measurement efforts, the completeness of the reanalysis and the detailed quantification of uncertainties are appreciable.*

*Due to logistical issues and/or to the peculiar characteristics of monitored glaciers, it is often impossible to setup point monitoring networks that are evenly distributed and cover the entire surface of the glaciers. Therefore, there is usually the need for interpolating and extrapolating point measurements over unmeasured areas, which often have high lateral gradients of mass balance. In such circumstances, the analyst's knowledge of the monitored glacier becomes decisive, and greatly benefit from repeat observations of snow cover patterns during the ablation season, over several years. The Authors of this paper make extensive use of this information for mapping the mass balance distribution over Langenferner, which in my opinion is another added value of this work.*

*However, I have one point to highlight, which is the calculation of so-called pseudo-observations in the upper part of the glacier for years without direct measurements in that area. Galos et al. use a physically based energy and mass balance model to do that, starting from meteorological data recorded by weather stations located in the proximity of the glacier. There are some weak points on the way the model has been applied and the transfer function for meteorological variables have been calculated (see detailed comments); apart from this, my principal criticism concerns the use of mass balance models for calculating artificial point measurements, which are required in usampled areas for completing glacier-wide mass balance computations.*

*Because the high value of mass balance records lies, among others, in their useability as sensitive climatic indicators and for improved understanding of glacier-climate interactions, using pseudo-measurements modelled from meteorological data is a sort of circular procedure.*

*Previous works by e.g. Haefeli (1962); Jansson (1999); Carturan et al. (2009), and Kuhn et al. (2009) proposed extrapolation procedures that are independent from meteorological observations. In my paper on the mass balance series of the neighbouring La Mare Glacier (Carturan, 2016), I tried implementing these procedures for mass balance extrapolation, combining point measurements and snow cover pattern observations.*

*Therefore, I recommend at least mentioning these works in the Introduction section, clearly stating why a mass balance model has been preferred for mass balance calculations in unsampled areas, and discussing the limitations of this approach. In particular, I suggest discussing the generalizability of the method (e.g. for glaciers with few or absent meteorological data, indispensable for calculating transfer functions of meteorological variables, glacier cooling effect, etc), and the fields of application of mass balance series that are partly derived from meteorological data (ok e.g. for estimating contribution to sea level rise and management of regional water supplies, but maybe less reliable for early-detection strategies of climate change and process understanding).*

The author of the short-comment addresses points which have already been raised by the referees. Those are:

- the use of point mass balances modeled based on atmospheric input

- alleged weak-points in the application of the used model

- a missing discussion of previous works using statistical approaches to estimate the mass balance in unmeasured areas and

- a better motivation why the presented approach was used.

We address all those points in the detailed statements in section 1 and apart from that, we reply to the short-comment author's detailed comments below.

**4.2 Specific Comments**

*Page 2, Line 15-25: In the Introduction, I suggest mentioning published works dealing with point mass balance extrapolation methods that are independent from meteorological data (e.g. mass balance vs. altitude). See references in the general comments.*
We included a number of the presented references in an enhanced motivation discussing the choice of our approach in favour to other methods.

*Page 4, line 15: extensive net accumulation measurement. Also in the following I suggest specifying net accumulation where required.*
We changed the term to "annual accumulation" which is defined in (Cogley et al., 2011).

*Page 4, line 26: I suggest writing here which time system is used (fixed date)*
We did not follow this suggestion since annual mass balance where reported as fixed-date, but seasonal balances before the reanalysis were floating-date balances actually. The point is addressed in detail in section 3 of the manuscript.

*Page 5, line 26-29: did the Authors consider the possibility of using the bedrock topography, as obtained from geophysical measurements, instead of the surface topography, for identifying the divides?*
The GPR-campaign mentioned in the text of the paper allowed for an estimate of total glacier volume, but the spatial resolution in the upper glacier part is not suited to derive the glacier bed in a sufficiently accurate matter.

*Page 6, line 21: how did the Authors account for this problem? Here and in other parts of the paper the reader is left (temporarily) without explanations*
We inserted a cross reference to manuscript section 3.13 where the corresponding method is explained in detail.

*Page 7, line 3: stratigraphic correction only for snow cover (is it fresh snow after measurement?) and not for ablation? How was it done?*
We changed the text of the paragraph to make the purpose of this correction more obvious. Melt (and other) ablation was neglected since it must only be accounted for if the melt water

drains from the glacier during the correction period (i.e. between April $30^{th}$ and the date of winter balance measurement in early May). This was never observed at Langenferner at that time of the year. Snow pits dug for winter balance measurements indicate that the snow pack was never saturated with melt water, except for the extraordinary year 2007 when the snow pack in the lower glacier sections was saturated due to very warm weather in April. But even in that year conditions in early May were colder again leading to the assumption that even in 2007 melt water run-off can be neglected in this regard. Note that the applied correction method only corrects for the date difference between the end of the hydrological winter (April $30^{th}$ and the date of measurement). The reason for larger delays in measurements has always been unfavourable weather which in early May at altitudes of more than 2700 m (terminus of Langenferner) is commonly related to snow fall rather than significant melt.

We added an explanatory sentence to the respective subsection in the paper and referred the reader to the supplementary material where we explain the assumption and resulting restrictions in the transferability of the method.

*Page 7, line 12: raw precipitation or (gauge-undercatch) bias-corrected precipitation? Possible impact on calculations?*

Unless not explicitly stated otherwise, "precipitation" refers to raw measurements. Hardly any impact can be expected on mass balance calculations since the precipitation measurements are taken at a relatively wind sheltered location in a valley at 1851 m a.s.l. Furthermore, the model approach fits accumulation to observations for each year and location individually. Measurement errors would hence impact precipitation scaling factors, rather than the mass balance results which are in any case validated against observations.

*Page 7, line 29-31: it is a fact that the spatial variability of the mass balance is large, and largely dependent on the spatial variability of micro-meteorological variables and local topography. Does the model fully account for these factors? Maybe the Authors should briefly recall here the main processes accounted for in the model. Moreover, physically based energy and mass balance models strongly rely on accurate spatially distributed fields (or in situ measurements) of several meteorological variables for their application. Because this is not the case of Langenferner, as stated at line 27 and in Section 3.2.1, there is the need for spatially flexible model tuning for the choice of the optimal parameters at the individual points, based on summer snowline observations. In my opinion, such use of a physically-based energy and mass balance model, instead of simple statistical relationships applied to measured point mass balances and observed snow lines (i.e. independent from meteorological observations, e.g. Carturan, 2016) is questionable and should be better motivated by the Authors.*

The applied method accounts for spatial and temporal variations of the most important mass balance drivers: short and long wave radiation, sensible heat flux and accumulation, as well as feedbacks related to it (surface albedo, etc) to which we devote special attention in our study by the flexible tuning of the precipitation scaling factor $\Gamma$.

Again we have to mention that our study does not aim to resolve physical processes behind glacier energy and mass balance but at simulating the mass balance at a given point as realistically as possible. Certainly accurate micro-meteorological fields as model input are of great importance when the goal is an enhanced understanding of physical processes. This issue is addressed, using Langenferner as the case study, in a recently published paper (Sauter and Galos, 2016). But for this study such problems are of minor importance for the reasons mentioned above.

Nevertheless, we included an enhanced motivation and discussion in the introduction of the revised paper.

*Page 7, line 31 to Page 8, Line 2: I think that actual measurements, instead of pseudo-observations derived from climatic/meteorological data, should be preferably used in investigations concerning glacier-climate interactions (there is a risk of circular reasoning).*
Please see our statements in section 1 of this document.

*Page 8, line 7-8: fixed lapse rates and fixed offsets of temperature imply fixed glacier cooling/damping effects and simple air temperature distribution, whereas in the recent literature there is evidence of a significant variability of these processes within/among glaciers (e.g. Greuell and Böhm, 1998; Shea and Moore, 2010; Petersen and Pellicciotti, 2011; Petersen et al., 2013; Carturan et al., 2015). The same is valid for the other meteorological variables, which have been calculated using transfer functions derived at one point over the glacier. I suggest at least discussing this issue.*

We apply the fixed temperature lapse rate which was also used by Carturan et al. (2012) and many other studies which aim to enhance process understanding related to glacier mass balance. In our study the applied temperature gradient is justified by on-glacier measurements at a weather station close to stake 22 for which the Monte Carlo optimization was performed. Besides that, we repeat that our study does not focus on individual processes contributing to the mass balance. Please also see our detailed statements in section 1 of this document and our replies to a series of similar comments above.

*Page 8, line 24: I think the main reason for this improvement lies on a better calculation of the liquid vs. solid precipitation fractions. Do the Authors agree?*
Not only. This also for example allows a snow pack to build up during favourable conditions in a certain period of the day, while the small amounts of equally distributed hourly snow fall would eventually melt during the same time step. The consequence is a change in albedo and other feedbacks which can be decisive for whether or not melt conditions occur.

*Page 8, line 28: $\Gamma_0$ therefore mainly accounts for gauge under-catch errors, precipitation vertical and horizontal gradients, and snow redistribution (and possibly ablation?). I suggest commenting on this.*
It accounts for the local accumulation characteristics which is explained in the following paper section 3.2.3. Please also see our above comment referring to Page 7, line 29-31.

*Page 9, line 16-18: this tuning procedure is based on the assumption that the parameters controlling ablation processes, tuned at stake 22, are also valid elsewhere and that there are no significant biases in the calculation of spatial fields of meteorological variables, because otherwise there could be compensating effects from errors in modelling accumulation and ablation processes. My suggestion is to briefly discuss this aspect. In addition, I suggest better clarifying and repeating here for which stakes and in which years the model calculations have been done.*

The tuning procedure is based on the assumption that the model setting after the Monte Carlo optimization is valid for the surrounding stakes which are all located in the upper half of the glacier at similar altitudes (approximately 3125 to 3360 m a.s.l., stake 22: 3220 m a.s.l). This does not mean that the physical processes governing the mass balance have to be the same since the model resolves local topography and its impact on insolation, altitudinal variations in air temperature and related feedbacks with long wave radiation and - especially important - local characteristics of accumulation. The set of 33 observations which is used to validate the approach comprises measurements of annual point balances ranging from -1900 $kg\ m^{-2}$ to slightly positive values and there is no detectable dependence of the model skill to the actual value of measured point balance. Even if there were compensating effects in the modeling of ablation and accumulation processes - which is so far nothing but an unqualified suspicion - it would not matter since this study does not aim at studying physical processes and the only interest is the skill of the modeled mass balance values.

The revised paper version includes a stronger motivation regarding the chosen approach including a discussion of alternatives. We have also tried to more clearly delineate the present study from works which focus on enhanced understanding of mass and energy balance processes. We clearly discuss the transferability of the model and its limitations in the introduction, as well as in section 5 (Results and Discussion). The number of modeled point values used in the calculations of glacier wide balances for individual years and seasons is now visible in a revised Figure 2 and an improved Table 1.

*Page 15, line 31: is this assessment bases on the procedure proposed by Rolstad et al. (2009)? If so, I suggest adding this reference here.*
Is is not explicitly based on Rolstad et al. (2009) but rather follows Joerg et al. (2012).
The reference was added.

*Page 16, line 5. Compared to the period from 2005 to 2012, in 2013 the AAR was significantly larger in most glaciers of the Ortles-Cevedale. The effects on the mean glacier density were likely small or negligible, but maybe the Authors could shortly comment on that.*
This issue is mentioned in paper section 3.4.
The effects are not negligible for short periods (2011 to 2013) which is now stated in section 5.4 of the revised manuscript.

*Page 17, line 26: maybe here the Authors could replace objective with automatic extrapolations. Here and elsewhere, specify net (accumulation) where required.*
Done.

*Page 19, line 23-24: in my opinion this is a key point and an essential prerequisite for reliable mass balance mapping and calculations. The Authors should mention here previous works highlighting the importance of observations regarding the snow cover pattern (e.g. Kaser et al., 2003; Østrem and Brugman, 1991).*
Done.

*Page 21, line 17-18: this good agreement is largely dependent on snow line observations, which have been used as constrains for model calibration at individual points, and on the representative location*

*of stake 22 (where the model has been optimized) compared to the location of the modelled stakes. Given also the simplified transfer functions used for the meteorological variables, I wonder how much generalizable the proposed method is, what is its added value in comparison to statistical procedures based only on observations (point measurements and snow line mapping), and how to consider mass balance pseudo-observations derived from climatic data used alongside actual observations. In my opinion, the implementation of such combined (measured + modelled from meteorological data) mass balance datasets poses some limitations to their use for process-understanding or as key variables in monitoring strategies of the Earth climate (Zemp and others, 2005; WGMS, 2008). This is because the climatic indicator (glacier mass balance) becomes dependent on the same climatic variables it should be a proxy of.*

Again please see our comments in the detailed statements in section 1 of this document and our replies to a series of similar comments above.

*M, Frauenfelder R, Haeberli W and Hoelzle M (2005) Worldwide glacier mass balance measurements: general trends and first results of the extraordinary year 2003 in Central Europe. In Data of Glaciological Studies [Materialy glyatsiologicheskih issledovanii], Moscow, Russia, vol. 99, 3–12.*

*(2008) In Zemp M and 5 others eds. Global glacier changes: facts and figures. UNEP, World Glacier Monitoring Service, Zürich, Switzerland*

*Figure 2: it is unclear if the snow probings refer to winter mass balance measurements or to annual net accumulation measurements. In the second case, the location of winter balance measurements should be added.*

The figure caption was changed for clarity.

*Figure 7: I suggest adding the 1:1 line*

Done.

**References**

Anonymous: Mass balance terms, Journal of Glaciology, 8, 3–7, 1969.

Barandun, M., Huss, M., Sold, L., Farinotti, D., Azisov, E., Salzmann, N., Usubaliev, R., Merkushkin, A., and Hoelzle, M.: Re-analysis of seasonal mass balance at Abramov glacier 1968-2014, Journal of Glaciology, 61, 1103–1117, doi:10.3189/2015JoG14J239, 2015.

Carturan, L.: Replacing monitored glaciers undergoing extinction: a new measurement series on La Mare Glacier (Ortles-Cevedale, Italy), journal of Glaciology, 236, 1093–1103, doi: 10.1017/jog.2016.107, 2016.

Carturan, L., Cazorzi, F., and Dalla Fontana, G.: Enhanced estimation of glacier mass balance in unsampled areas by means of topographic data, Annals of Glaciology, 50, 37–46, doi: doi:10.3189/172756409787769519, 2009.

Carturan, L., Cazorzi, F., and Dalla Fontana, G.: Distributed mass-balance modelling on two

neighbouring glaciers in Ortles-Cevedale, Italy, from 2004 to 2009, Journal of Glaciology, 58, 467–486, doi:10.3189/2012JoG11J111, 2012.

Carturan, L., Cazorzi, F., De Blasi, F., and Dalla Fontana, G.: Air temperature variability over three glaciers in the Ortles–Cevedale (Italian Alps): effects of glacier fragmentation, comparison of calculation methods, and impacts on mass balance modeling, The Cryosphere, 9, 1129–1146, doi:10.5194/tc-9-1129-2015, 2015.

Cogley, J. G., Hock, R., Rasmussen, L. A., Arendt, A. A., Bauder, A., Braithwaite, R. J., Jansson, P., Kaser, G., Möller, M., Nicholson, L., and Zemp, M.: Glossary of Glacier Mass Balance and Related Terms, IHP-VII Technical Documents in Hydrology No. 86, IACS Contribution No. 2, UNESCO-IHP, Paris, 2011.

Galos, S., Klug, C., Prinz, R., Rieg, L., Sailer, R., Dinale, R., and Kaser, G.: Recent glacier changes and related contribution potential to river discharge in the Vinschgau / Val Venosta, Italian Alps, Geografia Fisica e Dinamica Quaternaria, 38, 143–154, doi: 10.4461/GFDQ.2015.38.13, 2015.

Greuell, W. and Böhm, R.: 2 m temperatures along melting mid-latitude glaciers, and implications for the sensitivity of the mass balance to variations in temperature, Journal of Glaciology, 44, 9–20, 1998.

Haefeli, R.: The ablation gradient and the retreat of a glacier tongue., In: Syposium of Obergurgl, IASH Publication, 58, 49–59, 1962.

Huss, M., Bauder, A., and Funk, M.: Homogenization of long-term mass balance time series, Annals of Glaciology, 50, 198–206, doi:10.3189/172756409787769627, 2009.

Huss, M., Sold, L., Hoelzle, M., Stokvis, M., Salzmann, N., Farinotti, D., and Zemp, M.: Towards remote monitoring of sub-seasonal glacier mass balance, Annals of Glaciology, 54, 75–83, doi:10.3189/2013AoG63A427, 2013.

Huss, M., Dhulst, L., and Bauder, A.: New long-term mass-balance series for the Swiss Alps, Journal of Glaciology, 61, 551–562, doi:10.3189/2015JoG15J015, 2015.

Jansson, P.: Effect of uncertainties in measured variables on the calculated mass balance of Storglaciären, Geografiska Annaler: Series A, Physical Geography, 81, 633–642, doi: 10.1111/1468-0459.00091, 1999.

Joerg, P. C., Morsdorf, F., and Zemp, M.: Uncertainty assessment of multi-temporal airborne laser scanning data: A case study on an Alpine glacier, Remote Sensing of Environment, 127, 118–129, doi:10.1016/j.rse.2012.08.012, 2012.

Kaser, G., Fountain, A., and Jansson, P.: A manual for monitoring the mass balance of mountain glaciers by, IHPVI Technical documents in Hydrology, 2003, 135, URL `http://glaciers.pdx.edu/fountain/MyPapers/KaserEtAl2002.pdf`, 2003.

Kronenberg, M., Barandun, M., Hoelzle, M., Huss, M., Farinotti, D., Azisov, E., Usubaliev, R., Gafurov, A., Petrakov, D., and Kääb, A.: Mass-balance reconstruction for Glacier No. 354, Tien Shan, from 2003 to 2014, Annals of Glaciology, 57, 92–102, doi:10.3189/2016AoG71A032, 2016.

Kuhn, M.: Mass Budget Imbalances as Criterion for a Climatic Classification of Glaciers, Geografiska Annaler. Series A, Physical Geography, 66, 229–238, 1984.

Kuhn, M., Abermann, J., Bacher, M., and Olefs, M.: The transfer of mass-balance profiles to unmeasured glaciers, Annals of Glaciology, 50, 185–190, doi:doi:10.3189/172756409787769618, 2009.

Lliboutry, L.: Multivariate statistical analysis of glacier annual balances, Journal of Glaciology, 13, 371–392, 1974.

Machguth, H., Paul, F., Hoelzle, M., and Haeberli, W.: Distributed glacier mass-balance modelling as an important component of modern multi-level glacier monitoring, Annals of Glaciology, 43, 335–343, doi:10.3189/172756406781812285, 2006.

Petersen, L. and Pellicciotti, F.: Spatial and temporal variability of air temperature on a melting glacier: Atmospheric controls, extrapolation methods and their effect on melt modeling, Juncal Norte Glacier, Chile, Journal of Geophysical Research: Atmospheres, 116, doi:10.1029/2011JD015842, d23109, 2011.

Petersen, L., Pellicciotti, F., Juszak, I., Carenzo, M., and Brock, B.: Suitability of a constant air temperature lapse rate over an Alpine glacier: testing the Greuell and Böhm model as an alternative, Annals of Glaciology, 54, 120–130, 2013.

Rasmussen, L. A.: Altitude variation of glacier mass balance in Scandinavia, Geophysical Research Letters, 31, n/a–n/a, doi:10.1029/2004GL020273, l13401, 2004.

Rolstad, C., Haug, T., and Denby, B.: Spatially integrated geodetic glacier mass balance and its uncertainty based on geostatistical analysis: Application to the western Svartisen ice cap, Norway, Journal of Glaciology, 55, 666–680, doi:10.3189/002214309789470950, 2009.

Sauter, T. and Galos, S. P.: Effects of local advection on the spatial sensible heat flux variation on a mountain glacier, The Cryosphere, 10, 2887–2905, doi:10.5194/tc-10-2887-2016, 2016.

Shea, J. and Moore, R.: Prediction of spatially distributed regional-scale fields of air temperature and vapor pressure over mountain glaciers, Journal of Geophysical Research: Atmospheres, 115, 2010.

Sold, L., Huss, M., Machguth, H., Joerg, P. C., Leysinger Vieli, G., Linsbauer, A., Salzmann, N., Zemp, M., and Hoelzle, M.: Mass Balance Re-analysis of Findelengletscher, Switzerland; Benefits of Extensive Snow Accumulation Measurements, Frontiers in Earth Science, 4, doi:10.3389/feart.2016.00018, 2016.

Østrem, G. and Brugman, M.: Glacier Mass-Balance Measurements: A Manual for Field and Office Work, NHRI Science Report, Saskatoon, Canada, doi:10.2307/1551489, 1991.

Thibert, E. and Vincent, C.: Best possible estimation of mass balance combining glaciological and geodetic methods, Annals of Glaciology, 50, 112–118, doi:10.3189/172756409787769546, 2009.

Zemp, M., Thibert, E., Huss, M., Stumm, D., Rolstad Denby, C., Nuth, C., Nussbaumer, S. U., Moholdt, G., Mercer, A., Mayer, C., Joerg, P. C., Jansson, P., Hynek, B., Fischer, A., Escher-Vetter, H., Elvehøy, H., and Andreassen, L. M.: Reanalysing glacier mass balance measurement series, Cryosphere, 7, 1227–1245, doi:10.5194/tc-7-1227-2013, 2013.